# From Conflict to Consensus: Boosting Medical Reasoning via Multi-Round Agentic RAG

Wenhao Wu [1 2]  Zhentao Tang [2 ✉]  Yafu Li [3]  Shixiong Kai [2]  Mingxuan Yuan [2]
Zhenhong Sun [4]  Chunlin Chen [1]  Zhi Wang [1 ✉]

## Abstract

Large Language Models (LLMs) exhibit high reasoning capacity in medical question-answering, but their tendency to produce hallucinations and outdated knowledge poses critical risks in healthcare fields. While Retrieval-Augmented Generation (RAG) mitigates these issues, existing methods rely on noisy token-level signals and lack the multi-round refinement required for complex reasoning. In this paper, we propose **MA-RAG** (**M**ulti-Round **A**gentic RAG), a framework that facilitates test-time scaling for complex medical reasoning by iteratively evolving both external evidence and internal reasoning history within an agentic refinement loop. At each round, the agent transforms semantic **conflict** among candidate responses into actionable queries to retrieve external evidence, while optimizing history reasoning traces to mitigate long-context degradation. MA-RAG extends the *self-consistency* principle by leveraging the lack of consistency as a proactive signal for multi-round agentic reasoning and retrieval, and mirrors a *boosting* mechanism that iteratively minimizes the residual error toward a stable, high-fidelity medical **consensus**. Extensive evaluations across 7 medical Q&A benchmarks show that MA-RAG consistently surpasses competitive inference-time scaling and RAG baselines, delivering **substantial +6.8 points** on average accuracy over the backbone model. Our code is available at https://github.com/NJU-RL/MA-RAG.

[1]Nanjing University [2]Huawei Noah's Ark Lab [3]The Chinese University of Hong Kong [4]Australian National University. (Email: wenhaowu@smail.nju.edu.cn). ✉ Correspondence to: Zhi Wang <zhiwang@nju.edu.cn>, Zhentao Tang <tangzhentao1@huawei.com>.

*Proceedings of the 43rd International Conference on Machine Learning*, Seoul, South Korea. PMLR 306, 2026. Copyright 2026 by the author(s).

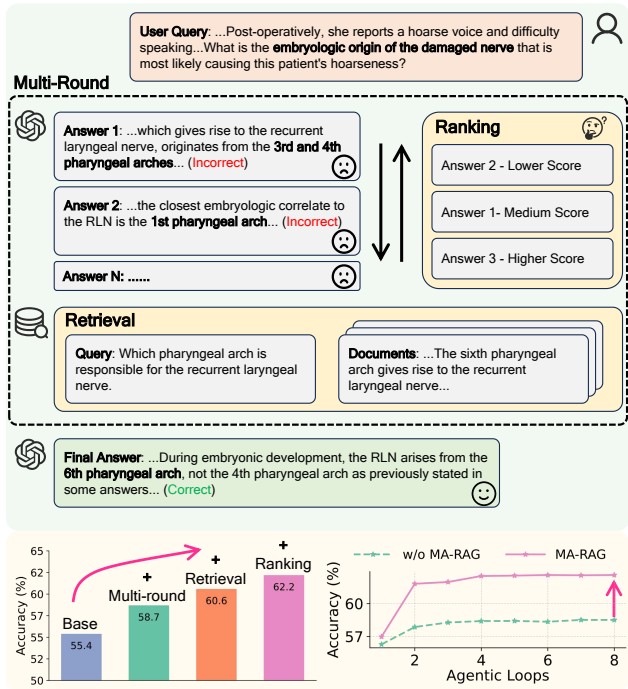

*Figure 1.* MA-RAG's agentic workflow that iteratively refines conflict into a unified consensus, achieving superior test-time scaling.

## 1. Introduction

Large Language Models (LLMs) (Hurst et al., 2024; Guo et al., 2025) have achieved substantial progress in language understanding and multi-step reasoning, yielding strong performance across a broad range of downstream tasks (Cui et al., 2025; Muennighoff et al., 2025; Yan et al., 2025; Hu et al., 2025; 2026; Zhan et al., 2026; Zhang et al., 2026). Building on these advances, a growing body of research adapts LLMs to the medical domain, yielding specialized medical models that hold promise for broad healthcare fields such as question-answering, decision support, and text understanding (Chen et al., 2024; Team et al., 2025; Sounack et al., 2025). Despite pretraining on massive corpora, medical LLMs remain prone to generating fluent yet factually incorrect *hallucinations* (Ji et al., 2023; Kalai et al., 2025), posing critical risks in safety-sensitive healthcare scenar-

ios (Li et al., 2025c). Meanwhile, parametric knowledge stored in model weights often becomes outdated, failing to align with emerging medical evidence or revised guidelines (Xiong et al., 2024).

Retrieval-Augmented Generation (RAG) has become a cornerstone paradigm that leverages external, verifiable medical evidence to provide up-to-date grounding necessary for factually accurate and reliable responses (Zhao et al., 2026; Yu et al., 2025). Traditional RAG often follows a single-round workflow: retrieving evidence based solely on an initial query before generating a final response conditioned on the retrieved context (Guu et al., 2020; Izacard et al., 2021). While effective for simple questions, one-shot retrieval often fails to provide evolving information required for complex, multi-step reasoning (Jiang et al., 2023; Su et al., 2024), and can introduce irrelevant noise to degrade the performance when retrieval is not universally beneficial (Choi et al., 2025). To tackle this, adaptive RAG methods (Jiang et al., 2023; Su et al., 2024; Jiang et al., 2025) interleave generation with multi-round retrieval by dynamically determining *when* to trigger external searches and *what* queries to issue. They usually rely on token-level signals, such as confidence (Jiang et al., 2023) or attention weights (Su et al., 2024), to construct context-aware queries. However, token-level uncertainty is often a poor proxy for retrieval needs, as LLMs may hallucinate with high confidence, and uncertainty estimates are frequently dominated by trivial words instead of domain-critical medical concepts required for precise query formulation. These limitations highlight a critical question: *Could we bypass the reliance on noisy token-level signals by leveraging higher-level semantic cues to steer agentic retrieval more efficiently?*

In medical domains, complex cases often elicit conflicting explanations or diagnoses when the model lacks sufficient evidence. The semantic **conflict** among multiple reasoning paths can provide a more grounded signal to identify where current knowledge is insufficient and retrieval augmentation is essential. By iteratively rectifying these conflicts through external evidence, the agentic loop acts as a refinement process that reconciles disparate reasoning paths into a reliable, high-fidelity **consensus**. This mechanism naturally aligns with principles of human cognitive science (Flavell, 1979), wherein individuals iteratively seek external validation or peer expertise to reduce inconsistencies.

Inspired by the above insights, we propose **MA-RAG** (**M**ulti-Round **A**gentic RAG), an agentic refinement process that steers test-time scaling for complex medical reasoning by iteratively evolving both external retrieval documents and internal reasoning history. Concretely, our pipeline consists of three agents: i) a **Solver Agent** that produces multiple candidate responses per round; ii) a **Retrieval Agent** that transforms semantic conflict among candidates into action-

able retrieval queries to seek external evidence from a local medical corpus; and iii) a **Ranking Agent** that optimizes history reasoning traces by prioritizing top-tier candidates, mitigating long-context degradation. MA-RAG extends the *self-consistency* principle by leveraging semantic inconsistency as a proactive signal for multi-round agentic reasoning and retrieval, and mirrors a *boosting* mechanism that iteratively minimizes residual errors (see Sec. 3.5).

Extensive experiments across seven medical Q&A benchmarks demonstrate MA-RAG's consistent superiority over competitive test-time-scaling and RAG baselines. Notably, MA-RAG achieves **substantial +6.8 points** on average accuracy over the backbone model. The performance gain is particularly pronounced on harder benchmarks (e.g., **a 37% relative improvement** over baselines on MedXpertQA) that require information-dense queries and sophisticated reasoning, highlighting our advantage in complex medical reasoning. Comprehensive analysis and case studies reveal that MA-RAG demonstrates robust performance during multi-round refinement and effectively steers test-time scaling.

## 2. Related Work

**RAG for Medical Reasoning.** RAG (Lewis et al., 2020) enhances LLMs by incorporating external knowledge, demonstrating significant potential for mitigating hallucinations in risk-sensitive medical domains (Xiong et al., 2024; Matsumoto et al., 2024; Zhao et al., 2026). Traditional *retrieval-and-generation* often retrieves redundant and noisy information, and struggles with complex, multi-hop reasoning tasks (Jiang et al., 2023; Su et al., 2024; Jiang et al., 2025). To address this, adaptive RAG dynamically determines *when* and *what* to retrieve. FLARE (Jiang et al., 2023) and DRA-GIN (Su et al., 2024) leverage internal signals, specifically low-confidence tokens or attention weights, to trigger retrieval. Other methods perform adaptive retrieval via query complexity classification (Jeong et al., 2024), state-managed processing (Jiang et al., 2025), or conflict mitigation by filtering or reconciling retrieved evidence (Choi et al., 2025; Zhang et al., 2025b; Wang et al., 2025). Despite these advancements, existing methods often rely on noisy token-level metrics (e.g., entropy), which may be unreliable due to the model's over-confidence (Kalai et al., 2025). In contrast, we use semantic conflict as a high-level, grounded signal to guide adaptive retrieval.

**Test-Time Scaling.** It emerges as a promising paradigm to enhance LLM reliability by allocating increasing compute during inference, consisting of *parallel scaling* and *sequential scaling*. Parallel scaling samples diverse reasoning paths independently and aggregates them to form a consensus using unsupervised mechanisms like majority voting (Wang et al., 2023; Chen et al., 2026). MedAdapter (Shi et al., 2024) and MedS$^3$ (Jiang et al., 2026) train specific veri-

fiers to select higher-quality responses in medical domains. Sequential scaling extends the reasoning to multi-round settings that utilize feedback from previous generations, achieving notable progress like OpenAI o1 (Jaech et al., 2024) and DeepSeek-R1 (Guo et al., 2025). Recent studies formulate generation as an iterative process of self-refinement (Tian et al., 2025; Xu et al., 2025; Li et al., 2025b;a), prompting models to critique and revise their own outputs (Madaan et al., 2023; Shinn et al., 2023). Based on this sequential paradigm, we propose an agentic refinement process to facilitate test-time scaling for complex medical reasoning.

## 3. Method

In this section, we present MA-RAG in detail, focusing on the agentic refinement process comprising the solver, retrieval, and ranking agents. Figure 2 illustrates the overall pipeline, and Appendix A shows the pseudocode.

### 3.1. Problem Statement

We formulate our multi-round adaptive RAG process as an *iterative context optimization* problem (Mei et al., 2025). The LLM's prompt is treated as a dynamic, evolving structure rather than a static sequence, allowing the context to adaptively mature over successive rounds. Let $\mathcal{M}$ and $\mathcal{A}_t$ denote the LLM and the set of generated responses at round $t \in \{1, \ldots, T\}$. For a given query $q$, the input state $\mathcal{S}_t$ at the $t$-th round is defined as a composite tuple:

$$\mathcal{S}_t = \{\mathcal{I}, q, \mathcal{D}_t, \mathcal{H}_t\}, \tag{1}$$

which contains the following components:

- $\mathcal{I}$ represents the *Task Instruction*, which is invariant.

- $\mathcal{D}_t$ denotes the *Document Context*, a dynamic set of medical passages retrieved to ground the model's reasoning. Unlike static single-round retrieval, $\mathcal{D}_t$ evolves through successive rounds, continuously updating the initial evidence pool with newly retrieved data.

- $\mathcal{H}_t = \text{Rank}(\mathcal{A}_{t-1})$ is the *History Context*, a collection of candidate responses in the previous round. Rather than a simple concatenation of prior responses, $\mathcal{H}_t$ is a structured repository that is strategically ranked and organized to maximize its utility as a sequence of high-quality in-context demonstrations (refer to Sec. 3.4).

Central to our agentic refinement loop is how to effectively transition state $\mathcal{S}_t$ to the new round $\mathcal{S}_{t+1}$, a process designed to iteratively optimize the context toward eliciting a robust, high-quality answer. Specifically, we utilize the semantic conflict within $\mathcal{A}_t$ to guide the retrieval of $\mathcal{D}_{t+1}$ (i.e., resolving *what* to retrieve) and organize the structure of $\mathcal{H}_{t+1}$ (i.e., determining *how* to present history) to elicit higher-quality responses in the subsequent round. This mirrors a

**boosting** mechanism (Schapire, 1990; Chen et al., 2016) that compels each round to focus on "hard cases" and rectify prior residual errors through retrieving external evidence. Sec. 3.5 elaborates our theoretical grounding in the principle of classical boosting algorithms.

### 3.2. Solver Agent

The solver agent acts as the primary reasoning engine, navigating the solution space while identifying latent uncertainties within the model's internal knowledge. At round $t$, conditioned on the current state $\mathcal{S}_t$, the agent performs stochastic exploration of the solution space via temperature-controlled sampling and generates a diverse set of $N$ candidate responses $\mathcal{A}_t$ as

$$\mathcal{A}_t = \{a_t^1, a_t^2, \ldots, a_t^N\} \sim \mathcal{M}(\mathcal{I}_{\text{solver}}, q, \mathcal{D}_t, \mathcal{H}_t). \tag{2}$$

The diversity within candidate generations $\mathcal{A}_t$ is fundamental to our framework, grounded in the empirical insight that accurate reasoning chains tend to converge toward a stable consensus, while hallucinations often exhibit divergent inconsistencies (Manakul et al., 2023; Chen et al., 2026). This observation naturally aligns with the core principle of **self-consistency** decoding (Wang et al., 2023) that complex reasoning tasks typically admit multiple reasoning paths toward a correct consensus, which is elaborated in Sec. 3.5.

Leveraging this signal, subsequent agents formulate conflict-guided queries for directed retrieval and rank history reasoning traces to enhance in-context learning, boosting the solver agent toward progressively higher-fidelity responses. When yielding consistent responses, we denote this as reasoning convergence and terminate the agentic refinement loop, aggregating the consensus to produce the final answer.

### 3.3. Retrieval Agent

Recent adaptive RAG approaches leverage internal signals like token-level uncertainty for dynamic retrieval (Jiang et al., 2023; Su et al., 2024); however, such metrics are often undermined by the tendency of LLMs to generate hallucinations with over-confidence (Kalai et al., 2025). In medical domains, complex cases usually induce conflicting diagnoses when the model lacks sufficient evidence. The semantic conflict among multiple reasoning paths can serve as a more reliable diagnostic, pinpointing knowledge gaps where retrieval augmentation is most critical. This aligns with human intelligence, wherein people will iteratively seek external evidence or peer expertise to reconcile inconsistencies, especially in risk-sensitive healthcare fields.

Based on this insight, we propose a retrieval agent that leverages semantic conflict within the candidate set $\mathcal{A}_t$ as a reliable indicator for the model's current knowledge gap, and transforms that signal into actionable queries to efficiently retrieve external evidence. The underlying premise

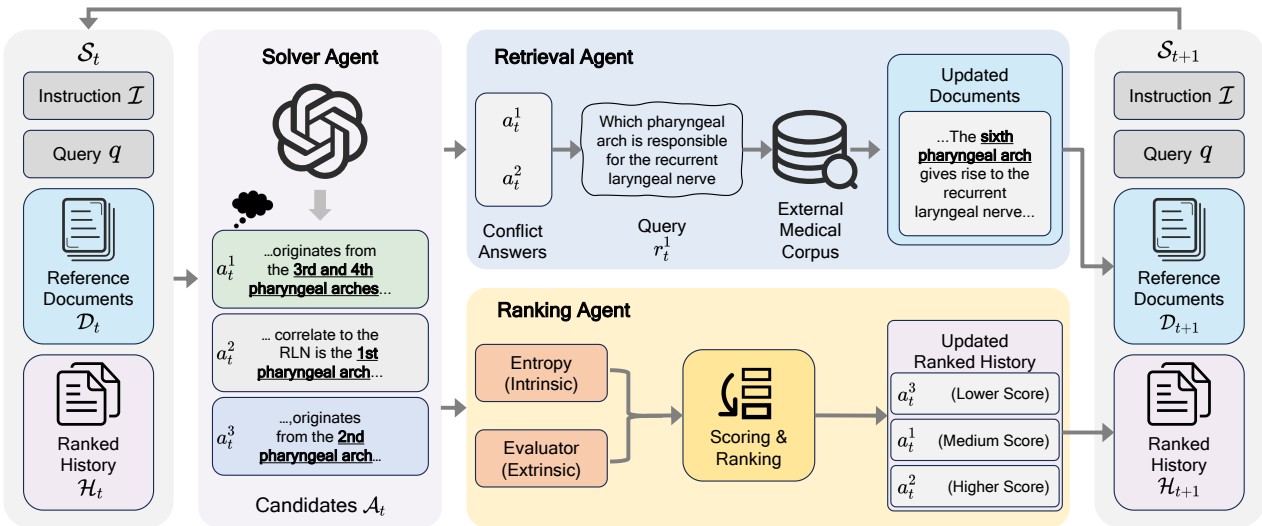

*Figure 2.* Overall pipeline of **MA-RAG** (**M**ulti-round **A**gentic RAG) for complex medical reasoning. At each round $t$ of the agentic refinement loop: i) the **Solver Agent** first samples a diverse set of candidate responses; ii) the **Retrieval Agent** transforms semantic conflict among candidates into actionable queries to retrieve external evidence from a local medical corpus, updating the document context to $\mathcal{D}_{t+1}$; and iii) the **Ranking Agent** restructures history reasoning traces $\mathcal{A}_t$ by prioritizing top-tier candidates to construct the history context $\mathcal{H}_{t+1}$, mitigating long-context degradation and enhancing in-context learning. The evolved state $S_t = \{\mathcal{I}, q, \mathcal{D}_t, \mathcal{H}_t\}$ serves as the prompt at the next round, iteratively rectifying semantic **conflict** toward converging to a reliable, high-fidelity **consensus**.

is that sufficient knowledge and reasoning capacity lead to *self-consistency* across multiple independent generations, whereas conflict signals a critical deficiency in grounded evidence. At each round, the agent extracts candidate conflicts (e.g., inconsistent diagnoses or symptom interpretations) as $\mathcal{I}_{\text{conflict}}$, followed by formulating a set of $K$ targeted retrieval queries $\mathcal{R}_t$ aimed to rectify these specific conflicts:

$$\mathcal{R}_t = \{r_t^1, r_t^2, \ldots, r_t^K\} \sim \mathcal{M}(\mathcal{I}_{\text{conflict}}, q, \mathcal{A}_t). \quad (3)$$

Then, the agent executes these conflict-aware queries against the external medical corpus to fetch new evidence $\mathcal{D}_{t+1}$, progressively narrowing the knowledge gap and providing the solver agent with augmented evidence required to rectify previous inconsistencies in subsequent rounds.

### 3.4. Ranking Agent

A significant bottleneck in sequential test-time scaling is long-context degradation (Mei et al., 2025), specifically the "lost-in-the-middle" issue, where models may overlook critical reasoning cues situated centrally within an expanding prompt (Zhang et al., 2025a). The refinement process is not only about augmenting more external evidence, but also optimizing the context to enhance in-context learning.

Based on this insight, we propose a ranking agent to *restructure* history reasoning traces, functioning as a context optimizer. The ranking agent uses a score function $Q(\cdot)$ to evaluate the quality of candidate responses in the previous round $\mathcal{A}_{t-1}$, and constructs the history context $\mathcal{H}_t$ as

$$\mathcal{H}_t = \text{sort}\left(\mathcal{A}_{t-1}, \text{key} = Q\right) = \left(a_{t-1}^{(1)}, a_{t-1}^{(2)}, \ldots, a_{t-1}^{(N)}\right), \quad (4)$$

where the indices are reordered such that the quality scores follow a descending order as $Q(a_{t-1}^{(1)}) \geq Q(a_{t-1}^{(2)}) \geq \cdots \geq Q(a_{t-1}^{(N)})$. This ensures that promising reasoning traces serve as prioritized demonstrations, effectively mitigating the long-context degradation issue. We use two representative scoring functions as follows.

**Intrinsic Uncertainty.** As entropy often serves as a simple, cheap metric for estimating generation quality (Manakul et al., 2023; Jiang et al., 2025; Sharma et al., 2025), we compute the sequence-level entropy as the score function for evaluating each response $a \in \mathcal{A}_{t-1}$:

$$Q_{\text{int}}(a) = -\frac{1}{L} \sum_{i=1}^{L} \text{entropy}\left(P(x_i \mid x_{<i}, S_t)\right), \quad (5)$$

where $x_i$ is the $i$-th token in response $a$, $x_{<i}$ is the sequence before $x_i$, and $L$ is the length of response $a$.

**Extrinsic Verification.** Beyond the token-level statistics, we design a higher-level score function that uses an auxiliary verifier to capture semantic correctness, specifically a lightweight BERT-based evaluator $V_\theta$. The model is fine-tuned as a binary classifier on a dataset of query-response pairs $\mathcal{D}_{qa} = \{(q_i, a_i)\}_{i=1}^{M}$ using a cross-entropy loss:

$$\mathcal{L}(\theta) = -\frac{1}{M} \sum_{i=1}^{M} y_i \log(s_i) + (1 - y_i) \log(1 - s_i), \quad (6)$$

where $y_i$ is the ground truth, and $s_i = \sigma(V_\theta(q_i, a_i))$ is the predicted probability. During inference, we use the fine-tuned verifier to compute the score function for $a \in \mathcal{A}_{t-1}$:

$$Q_{\text{ext}}(a) = V_\theta(q, a). \tag{7}$$

Appendix E presents details of constructing the dataset $\mathcal{D}_{qa}$ and fine-tuning the external verifier.

### 3.5. Theoretical Grounding in Classic Principles

**From Static to Adaptive Self-Consistency.** Self-consistency (Wang et al., 2023) is a simple, classical decoding strategy that achieves a striking margin on chain-of-thought reasoning. Standard self-consistency simply takes a majority vote to select the most consistent answer, assuming the model's internal knowledge is sufficient to reach a consensus in a single round. The mathematical objective is:

$$\max_a \sum_{i=1}^N \frac{1}{N} \cdot \mathbb{1}(a_i = a). \tag{8}$$

MA-RAG extends this principle by leveraging semantic "inconsistency" as a signal to keep thinking and retrieving in a multi-round setting. While MA-RAG optimizes the same objective in Eq. 8, it introduces a confidence threshold $\epsilon$ as a gating mechanism. If the maximum consistency probability falls below $\epsilon$, the agent triggers external retrieval to rectify the conflict in the next round. This transforms static self-consistency into *an adaptive scaling mechanism*: if current responses do not converge to a stable consensus, the agent triggers an additional round of retrieval, efficiently scaling test-time compute only when needed.

**Semantic Conflict as a Boosting Residual.** Classical boosting algorithms (Friedman, 2001; Chen et al., 2016) train each successive weak learner to minimize the "residual error" left by the previous learners. Analogously, MA-RAG frames the semantic conflict identified in round $t$ as a "boosting residual", a knowledge gap that remains unsolvable given the current state $\mathcal{S}_t$. The solver agent diagnoses the gradient direction from the residuals found in candidate responses, while the retrieval and ranking agents provide the external evidence and contextual optimization required to "fit" this residual. By incorporating this feedback into state $\mathcal{S}_{t+1}$, MA-RAG performs sequential refinement of the reasoning path. This boosting mechanism iteratively minimizes the "loss" (semantic conflict) until the system converges to a strong-learner state characterized by a stable, high-fidelity consensus across all reasoning trajectories.

## 4. Experiments

In this section, we present a comprehensive evaluation of MA-RAG on a diverse set of seven medical reasoning benchmarks, aiming to answer the following research questions:

- Can MA-RAG achieve consistent superiority on diverse benchmarks compared to competitive test-time scaling and RAG baselines? (Sec. 4.1)

- Can conflict-guided retrieval pinpoint knowledge gaps to rectify inconsistencies, and can ranking-based context optimization enhance in-context learning? (Sec. 4.2)

- Can MA-RAG steer efficient test-time scaling w.r.t. refinement rounds $T$, candidate numbers $N$ (Sec. 4.3), and backbone model capabilities? (Sec. 4.4)

- How does MA-RAG compare against competitive baselines in terms of end-to-end inference efficiency? (Sec. 4.5)

**Datasets.** We evaluate on seven medical Q&A benchmarks: MedQA (USMLE) (Jin et al., 2021), MedMCQA (Pal et al., 2022), MedExpQA (EN) (Alonso et al., 2024), Medbullets (5 options) (Chen et al., 2025), NEJM (Katz et al., 2024), the medical subset of MMLU-Pro (Wang et al., 2024), and MedXpertQA (Text) (Zuo et al., 2025). These datasets cover a broad spectrum of difficulty levels, ranging from standard medical licensing examinations to complex, expert-level clinical reasoning requiring multi-step information augmentation. See Appendix B for more details.

**Baselines.** We compare MA-RAG to 13 baselines that cover 5 representative paradigms for medical reasoning:

- 4 Backbones: **Qwen3-8B** (Yang et al., 2025), **Llama-3.1-8B-Instruct** (Grattafiori et al., 2024), and specialized medical LLMs of **UltraMedical-3.1-8B** (Zhang et al., 2024) and **HuatuoGPT-o1-8B** (Chen et al., 2024);

- 3 Test-time scaling methods without retrieval: **CoT** (Chain-of-Thought) (Wei et al., 2022), **SC** (Self-Consistency) (Wang et al., 2023), and **Multi-Refine** (Tian et al., 2025; Xu et al., 2025);

- 3 Naive RAG methods: **SR-RAG** (Single-Round RAG) (Xiong et al., 2024), **FL-RAG** (Fixed-Length RAG) (Borgeaud et al., 2022; Ram et al., 2023), and **FS-RAG** (Fixed-Sentence RAG) (Trivedi et al., 2023);

- 2 Adaptive RAG methods: **FLARE** (Jiang et al., 2023) and **TC-RAG** (Jiang et al., 2025);

- 1 Multi-agent collaboration method: **MDAgents** (Kim et al., 2024).

We report **accuracy (%)** as the primary metric across all benchmarks. For all RAG-based methods, we use the same medical corpus MedCorp from MedRAG (Xiong et al., 2024), the same retriever BM25 (Robertson et al., 2009), and the same reranker MedCPT-Cross-Encoder (Jin et al., 2023), to ensure a strictly fair comparison. Appendix D presents more implementation details.

*Table 1.* Main results (Accuracy %) on seven medical benchmarks. The best results are highlighted in **bold**, and the second-best are underlined. MA-RAG-int/MA-RAG-ext denote our method using the intrinsic uncertainty/extrinsic verification in the ranking agent.

| Method | MedQA | MedMCQA | Medbullets | MMLU-Pro | NEJM | MedExpQA | MedXpertQA | Avg. |
|---|---|---|---|---|---|---|---|---|
| *General and Medical LLMs* | | | | | | | | |
| Qwen3-8B | 71.1 | 61.3 | 51.0 | 64.9 | 56.0 | 67.2 | 16.1 | 55.4 |
| Llama-3.1-8B | 68.1 | 58.1 | 47.1 | 58.0 | 50.5 | 67.2 | 12.4 | 51.6 |
| UltraMedical-3.1-8B | 69.6 | 56.5 | 54.5 | 48.5 | 41.5 | 66.4 | 13.1 | 50.0 |
| HuatuoGPT-o1-8B | 71.1 | 60.2 | 50.6 | 54.0 | 47.9 | 63.2 | 15.5 | 51.8 |
| *Test-Time Scaling Methods (Backbone: Qwen3-8B)* | | | | | | | | |
| CoT | 69.3 | 60.3 | 50.6 | 63.4 | 56.2 | 72.0 | 18.0 | 55.7 |
| SC | 73.3 | 62.4 | 51.9 | 66.5 | 56.6 | 70.4 | 15.7 | 56.7 |
| MDAgents | 72.1 | **67.5** | 52.2 | 65.7 | 57.1 | 74.4 | 18.2 | 58.2 |
| Multi-Refine | 74.7 | 63.6 | 55.8 | 69.1 | 58.0 | 73.6 | 16.3 | 58.7 |
| *RAG Methods (Backbone: Qwen3-8B)* | | | | | | | | |
| SR-RAG | 69.9 | 64.4 | 49.4 | 66.1 | 57.6 | 72.8 | 17.3 | 56.8 |
| FL-RAG | 69.6 | 63.3 | 52.3 | 64.1 | 57.2 | 69.6 | 17.0 | 56.2 |
| FS-RAG | 68.0 | 61.3 | 51.3 | 59.5 | 53.1 | 67.2 | 16.5 | 53.8 |
| FLARE | 72.7 | 61.7 | 51.9 | 62.2 | 55.3 | 71.2 | 17.7 | 56.1 |
| TC-RAG | 70.0 | 64.6 | 49.0 | 63.6 | 57.7 | 75.2 | 18.0 | 56.9 |
| **MA-RAG**-int | 77.0 | 67.1 | 57.1 | **70.9** | **60.8** | 72.8 | 21.2 | 61.0 |
| **MA-RAG**-ext | **77.1** | 67.2 | **59.1** | 70.7 | 60.5 | **78.4** | **22.2** | **62.2** |

## 4.1. Main Results

Table 1 summarizes the performance comparison across all datasets. Qwen3-8B achieves the best performance across all backbone models. Hence, we implement MA-RAG and other baselines using the Qwen3-8B backbone.

**Superiority over Test-Time Scaling Baselines.** SC (Self-Consistency) yields a modest improvement over the base model (+1.3 points) by aggregating consensus across multiple reasoning traces in a single round. Building upon the principle of iterative self-improvement, Multi-Refine further enhances performance by 3.3 points over the backbone by allowing the model to self-refine generations across multiple rounds. By adaptively assigning collaboration structures of expert teams for debating and refining, MDAgents outperforms the single-expert backbone by 2.8 points. These results show that scaling inference compute can increase medical reasoning capacity to some extent.

Figure 3 shows the test-time scaling performance between MA-RAG and the multi-round baseline (Multi-Refine). Multi-Refine quickly reaches a performance plateau, revealing the bottleneck of traditional test-time scaling methods that stems from the base model's knowledge deficiencies in medical domains. In contrast, MA-RAG achieves continuous and substantial gains throughout the iterative process. The performance gain is particularly pronounced on harder benchmarks, e.g., a 37% relative improvement on MedXpertQA, highlighting our advantage in complex medical reasoning problems. By bridging knowledge gaps via adaptively injecting external evidence, MA-RAG effectively transcends the limits of pure inference scaling methods.

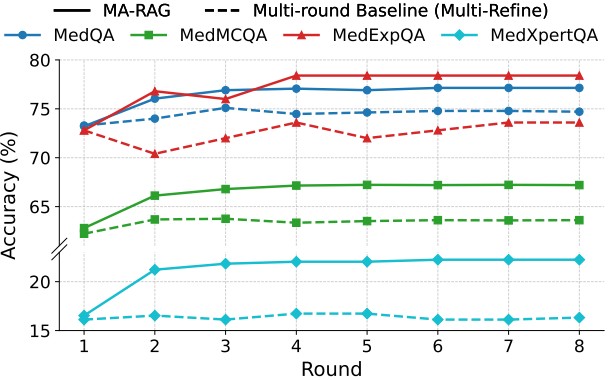

*Figure 3.* Performance comparison between MA-RAG and the multi-round test-time scaling baseline (Multi-Refine).

**Superiority over RAG methods.** Naive RAG methods often yield only marginal gains and may even suffer from performance degradation (e.g., on MedQA), compared to the Qwen3-8B backbone. This can stem from the insufficient or noisy context provided by single-round retrieval. While FLARE and TC-RAG employ multi-round retrieval, they also achieve only marginal gains, suggesting that simply increasing the retrieval rounds is insufficient for complex medical reasoning. MA-RAG addresses these pitfalls by employing the high-level semantic conflict as the retrieval signal, rather than the noisy token-level metrics used in adaptive RAG baselines. Notably, MA-RAG-ext achieves an average accuracy of 62.2%, significantly outperforming the strongest RAG baseline by a margin of 5.3 points.

**Score Functions in Ranking Agent.** MA-RAG-ext (using the BERT-based score function) yields an average gain of 1.2

*Table 2.* Ablation study on the components of MA-RAG using Qwen3-8B backbone. The best results are highlighted in **bold**.

| Method | MedQA | MedMCQA | Medbullets | MMLU-Pro | NEJM | MedExpQA | MedXpertQA | Avg. |
|---|---|---|---|---|---|---|---|---|
| Qwen3-8B | 71.1 | 61.3 | 51.0 | 64.9 | 56.0 | 67.2 | 16.1 | 55.4 |
| +Multi-Refine | 74.7 | 63.6 | 55.8 | 69.1 | 58.0 | 73.6 | 16.3 | 58.7 |
| +Retrieval Agent | **77.1** | 66.2 | 57.5 | 69.6 | 60.3 | 74.4 | 19.2 | 60.6 |
| +Ranking Agent | **77.1** | **67.2** | **59.1** | **70.7** | **60.5** | **78.4** | **22.2** | **62.2** |

points compared to MA-RAG-int (using the entropy-based score function). This observation aligns with our analysis of existing adaptive RAG methods: token-level uncertainty metrics may prove unreliable, as they fail to detect confident yet factually incorrect hallucinations. The external verifier, fine-tuned on specialized medical corpus, establishes a more robust mechanism for prioritizing high-quality demonstrations, leading to enhanced in-context learning. Appendix F.4 provides more quantitative analysis on the ranking agent.

### 4.2. Ablation Study

To disentangle respective contributions of individual components within MA-RAG, we conduct an ablation study using Qwen3-8B as the backbone. We employ a cumulative component-addition strategy, progressively building toward the full MA-RAG model through the following ablations:

- +Multi-Refine: implements iterative refinement based on internal knowledge, without external retrieval;

- +Retrieval Agent: integrates the conflict-guided retrieval agent to adaptively retrieve external evidence from a local medical corpus;

- +Ranking Agent: corresponds to the full MA-RAG method, which further incorporates the ranking mechanism to optimize history reasoning traces.

Table 2 shows the ablation performance on all benchmarks. +Multi-Refine achieves a substantial average gain of 3.3 points over the backbone, highlighting the significance of multi-round refinement for complex medical reasoning.

**Efficacy of the Retrieval Agent.** Including agentic retrieval (+Retrieval Agent vs. +Multi-Refine) yields an average gain of 1.9 points. Notably, these gains are more pronounced on knowledge-intensive benchmarks such as MedXpertQA, where the model's internal knowledge is often insufficient. When the model hallucinates due to a lack of factual grounding, repeated reasoning without external evidence typically fails to rectify the error. By leveraging semantic conflict as a diagnostic for knowledge gaps, our agent effectively retrieves targeted evidence, providing necessary context to ground the model toward the correct answer. In summary, our retrieval agent fulfills its objective of precise evidence acquisition, transcending the inherent performance ceilings of pure iterative self-refinement.

**Efficacy of the Ranking Agent.** In the absence of context

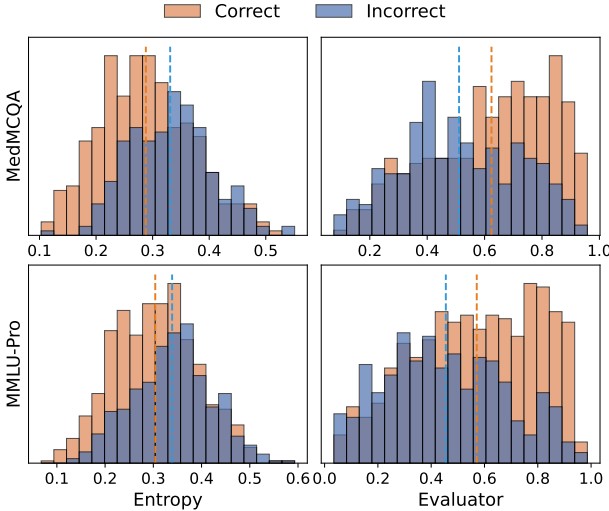

*Figure 4.* Visualization of the response score density on MedMCQA and MMLU-Pro. **Intrinsic Uncertainty** (left): Correct answers exhibit lower entropy compared to incorrect ones. **Extrinsic Verification** (right): The fine-tuned BERT-based evaluator assigns higher scores to correct answers, exhibiting a more pronounced discriminative margin than the entropy-based score. These distinct distributions validate the utility of both score functions for the ranking agent. Notably, the extrinsic evaluator exhibits superior discriminative power compared to the intrinsic counterpart, a finding consistent with the performance gap observed between MA-RAG-int and MA-RAG-ext in Table 1.

optimization (+Retrieval Agent), high-quality traces can suffer from the "lost-in-the-middle" issue, where critical evidence receives insufficient attention. When incorporating context optimization (+Ranking Agent), our method yields consistent performance gains, with +1.6 points on average and notable +4.0 points on MedExpQA. Figure 4 visualizes the response score density, confirming that both scoring functions provide a robust ranking mechanism for strategic reorganization of history reasoning traces. By explicitly prioritizing high-quality traces based on the scores, the ranking agent acts as a dynamic context optimizer, effectively mitigating long-context degradation and enhancing in-context learning performance.

### 4.3. Test-Time Scaling Analysis

To assess MA-RAG's test-time scaling properties, we investigate its performance regarding two core hyperparameters: the maximum refinement rounds $T$ and the size of the candidate pool $N$. Table 3 presents the results using Qwen3-8B.

*Table 3.* MA-RAG-ext's test-time scaling performance w.r.t. the maximum refinement rounds $T$ and the size of the candidate pool $N$.

|  | MedQA | MedMCQA | Medbullets | MMLU-Pro | NEJM | MedExpQA | MedXpertQA | Avg. |
|---|---|---|---|---|---|---|---|---|
| *Maximum Refinement Rounds $T$* | | | | | | | | |
| $T=1$ | 73.2 | 62.8 | 53.2 | 67.6 | 56.2 | 72.8 | 16.5 | 57.5 |
| $T=2$ | 76.0 | 66.1 | **60.4** | 69.8 | 60.2 | 76.8 | 21.2 | 61.5 |
| $T=4$ | **77.1** | **67.2** | 59.1 | **70.8** | 60.2 | **78.4** | 22.0 | 62.1 |
| $T=8$ | **77.1** | **67.2** | 59.1 | 70.7 | **60.5** | **78.4** | **22.2** | **62.2** |
| *Size of Candidate Pool $N$* | | | | | | | | |
| $N=2$ | 74.5 | 64.7 | 54.9 | 69.8 | 56.6 | 72.0 | 17.3 | 58.5 |
| $N=4$ | 76.0 | 65.7 | 58.4 | 68.2 | 58.8 | 74.4 | 18.6 | 60.0 |
| $N=8$ | **77.1** | **67.2** | **59.1** | **70.7** | **60.5** | **78.4** | **22.2** | **62.2** |

*Table 4.* Performance of MA-RAG's ablations based on the Qwen3-32B backbone, demonstrating its scalability across model capacities.

| Method | MedQA | MedMCQA | Medbullets | MMLU-Pro | NEJM | MedExpQA | MedXpertQA | Avg. |
|---|---|---|---|---|---|---|---|---|
| Qwen3-32B | 80.8 | 68.8 | 63.0 | 73.1 | 64.4 | 82.4 | 20.2 | 64.7 |
| +Multi-Refine | 83.6 | 70.9 | 64.9 | 75.5 | 68.4 | 86.4 | 21.2 | 67.3 |
| +Retrieval Agent | 83.5 | 72.8 | 70.8 | 76.3 | 69.2 | 86.4 | 24.9 | 69.1 |
| +Ranking Agent | **85.3** | **73.4** | **71.4** | **76.7** | **70.4** | **87.2** | **27.3** | **70.2** |

**Analysis of the Maximum Inference Rounds.** Notably, incorporating just a second round ($T=2$) yields a significant improvement of 4.0 points on average, demonstrating that even one additional round of conflict-guided retrieval is highly effective at bridging the model's knowledge gap. Performance gains begin to saturate beyond $T=4$, with the model exhibiting asymptotic behavior as it yields only a negligible point improvement when scaling to $T=8$. The scaling trend suggests that MA-RAG achieves effective and computationally efficient multi-round retrieval. Consequently, we posit that $T=4$ serves as a cost-effective stopping criterion for empirical deployment.

**Analysis of the Candidate Pool Size.** The diversity within the candidate pool is essential to MA-RAG, as it determines the capacity for both conflict-guided retrieval and rank-based context optimization. Expanding the pool size yields consistent performance gains, with 1.5 points when moving from $N=2$ to $N=4$, and another 2.2 points when scaling to $N=8$. We attribute this improvement to two factors: i) **Enhanced Conflict Mining**, a larger pool increases the coverage of semantic conflict, allowing the retrieval agent to formulate more precise queries for bridging knowledge gaps; and ii) **Improved In-Context Learning**, expanding the candidate pool enlarges the search space for high-quality reasoning traces, enabling the ranking agent to distill and prioritize superior in-context demonstrations. $N=4$ provides a cost-effective configuration for resource-constrained scenarios, and scaling to $N=8$ is preferable when maximizing reasoning performance is paramount.

**Scaling $N$ Delays Round-wise Saturation.** To further investigate the interaction between candidate diversity and multi-round refinement, we analyze the per-round perfor-

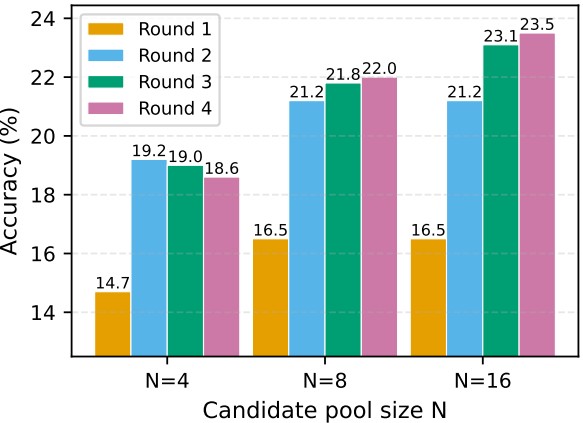

*Figure 5.* Per-round performance of MA-RAG-ext under varying candidate pool sizes $N$ on MedXpertQA with Qwen3-8B.

mance under varying pool sizes $N \in \{4, 8, 16\}$ on Medbullets and MedXpertQA. As shown in Figure 5 and Table 10, increasing $N$ yields two notable benefits. First, a larger pool of candidates improves peak performance, with $N=16$ delivering gains of 4.6 points on Medbullets and 4.3 points on MedXpertQA over $N=4$. Second, expanding $N$ defers the onset of saturation: while $N=4$ plateaus after Round 2, $N=16$ sustains meaningful improvements through Round 3 and beyond. This delayed saturation is attributed to the richer semantic conflict mined from a larger candidate pool, which in turn fuels more diverse and targeted retrieval queries across successive rounds.

### 4.4. Scalability across Model Scales

To verify MA-RAG's scalability to larger models, we evaluate its performance on Qwen3-32B backbone. Table 4

*Table 5.* Inference efficiency comparison across methods on MedXpertQA with Qwen3-8B. We report average retrieved documents, generated tokens, final answer tokens, and wall-clock time per question.

| Method | Avg. Docs | Avg. Gen. Tokens | Avg. Final Tokens | Avg. Time (s) | Accuracy |
|---|---|---|---|---|---|
| Base (Qwen3-8B) | 0 | 514 | 514 | 5.6 | 16.1 |
| Self-Consistency (SC) | 0 | 10,095 | 505 | 28.0 | 15.7 |
| FLARE | 30.0 | 962 | 429 | 42.7 | 17.7 |
| TC-RAG | 11.5 | 1,260 | 1,067 | 44.5 | 18.0 |
| MDAgents | 8.0 | 7,546 | – | 146.2 | 18.2 |
| MA-RAG-ext | 13.7 | 12,762 | 589 | 70.7 | **22.2** |

presents the ablation study on the Qwen3-32B model, and Figure 6 shows the performance comparison between MA-RAG and backbones across model scales. MA-RAG scales effectively with increasing model capacities, with a substantial average gain of 5.5 points when deployed on the stronger 32B backbone. Notably, the retrieval agent continues to play a critical role, boosting the performance with 1.8 points by rectifying knowledge gaps that remain unsolved even in larger models. The superiority is more evident on the challenging MedXpertQA benchmark, where the retrieval agent provides a substantial gain of 3.7 points and the ranking agent contributes an additional gain of 2.4 points. In summary, these results demonstrate MA-RAG's scalability across base model capacities, enabling consistent performance gains for complex medical reasoning tasks.

### 4.5. Inference Efficiency Analysis

To demonstrate the practical viability of multi-round agentic refinement, Table 5 reports a comprehensive efficiency analysis of MA-RAG-ext against representative baselines across retrieval, generation, and wall-clock time dimensions. All measurements are executed using the Qwen3-8B backbone on MedXpertQA under identical hardware conditions. Further results on Medbullets can be found in Appendix F.8.

**Retrieval Efficiency.** While MA-RAG-ext retrieves slightly more documents than TC-RAG, it yields a vastly superior performance improvement of 4.2 points. Furthermore, MA-RAG-ext requires less than half the retrieved documents of FLARE yet achieves a substantial 4.5 point gain in accuracy. This demonstrates that semantic conflict provides a more grounded and reliable retrieval signal for pinpointing knowledge gaps, in contrast to the noisy token-level uncertainty employed by existing adaptive RAG baselines.

**Generation Efficiency.** While MA-RAG-ext generates approximately $1.2\times$ more tokens than Self-Consistency, it delivers a substantial 6.5 point accuracy gain, demonstrating that its generation budget is productively invested in retrieval-augmented refinement rather than solely resampling reasoning paths without external grounding.

**Wall-Clock Time.** MA-RAG-ext outperforms the strongest baseline MDAgents by 4.0 points while requiring less than

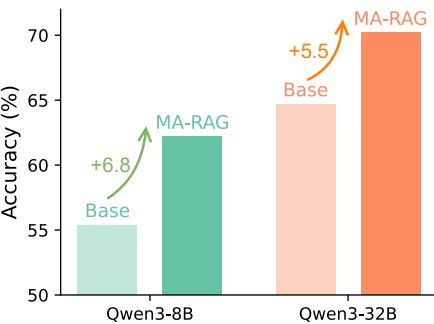

*Figure 6.* Performance of MA-RAG across backbone model scales.

half the inference time. Although it incurs approximately $1.6\times$ the time cost of TC-RAG, this yields a 4.2 point accuracy gain, representing a relative improvement of 23.3%. This demonstrates that MA-RAG-ext achieves a highly favorable cost-performance trade-off: a moderate increase at inference time for a rigorously refined, evidence-grounded diagnostic assistant.

## 5. Conclusions, Limitations, and Future Work

In this paper, we proposed MA-RAG, a novel framework that iteratively evolves both external evidence and internal reasoning history within an agentic refinement loop to steer test-time scaling for complex medical reasoning. At each round, the solver agent samples multiple candidate responses, the retrieval agent transforms semantic conflict among candidates into actionable queries to retrieve external evidence, and the ranking agent optimizes history reasoning traces to enhance in-context learning. Extensive experiments verified MA-RAG's consistent superiority over a variety of test-time scaling and RAG baselines.

Nevertheless, the inference-time cost of multi-round agentic refinement remains a limitation, and retrieval effectiveness is inherently constrained by the coverage and quality of the underlying medical corpus. Future work may extend MA-RAG to a more general agentic paradigm by incorporating richer tools such as web-scale search, structured medical databases, or external reasoning modules to broaden evidence access. Developing more reliable response quality estimation methods is also crucial, as the ranking agent's potential is inherently tied to evaluation accuracy.

## Acknowledgements

This work was supported in part by the National Natural Science Foundation of China under Grant 62376122, in part by the National Key Research and Development Program of China under Grant 2025YFA1016904, and in part by the National Natural Science Foundation of China under Grant 72394363.

## Impact Statement

The broader impact of MA-RAG lies in enabling more reliable and scalable large language model—powered AI systems for medical reasoning by introducing a multi-round agentic refinement framework that actively resolves uncertainty through iterative evidence retrieval and reasoning optimization. We anticipate MA-RAG will serve as a foundational component for future clinical AI systems, facilitating safer deployment of large language models and promoting evidence-grounded medical intelligence in real-world healthcare settings. However, despite its promising potential, the deployment of MA-RAG in real-world medical settings also raises some practical considerations. The framework relies on external retrieval sources, which may contain incomplete, outdated, or institution-specific medical knowledge, potentially introducing systematic biases into the reasoning process. It is also hard to fully eliminate hallucinations and guarantee the factual correctness of model-generated responses, even with iterative retrieval and reasoning refinement.

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

## A. Algorithm Pseudocodes

Based on the implementations in Sec. 3, this section gives the brief procedure of the MA-RAG framework in Algorithm 1. The process outlines the interaction between the solver, retrieval, and ranking agents across iterative refinement rounds.

---

**Algorithm 1** MA-RAG Workflow

---

**Require:** Large language model $\mathcal{M}$;  Instructions $\mathcal{I}_{\text{solver}}$ and $\mathcal{I}_{\text{conflict}}$;  Medical corpus $\mathcal{K}$;  Score function $Q$
**Require:** Medical query $q$
**Ensure:** Final answer $a$
 1: Initialize document context $\mathcal{D}_1 \leftarrow \emptyset$ and history context $\mathcal{H}_1 \leftarrow \emptyset$
 2: **for** $t = 1, \ldots, T$ **do**
 3:     // ▼ **Solver Agent**
 4:     Sample candidate responses $\mathcal{A}_t = \{a_t^1, \ldots, a_t^N\} \sim \mathcal{M}(\mathcal{I}_{\text{solver}}, q, \mathcal{D}_t, \mathcal{H}_t)$
 5:     **if** all candidates in $\mathcal{A}_t$ reach an identical conclusion **then**
 6:         **Terminate** the iterative process
 7:     **end if**
 8:     // ▼ **Retrieval Agent**
 9:     Initialize document context $\mathcal{D}_{t+1} \leftarrow \emptyset$
10:     Generate conflict-aware retrieval queries $\mathcal{R}_t = \{r_t^1, \ldots, r_t^K\} \sim \mathcal{M}(\mathcal{I}_{\text{conflict}}, q, \mathcal{A}_t)$
11:     **for** each query $r_t^k \in \mathcal{R}_t$ **do**
12:         Retrieve relevant documents $D_t^k \leftarrow \text{Retrieve}(r_t^k, \mathcal{K})$
13:         Update document context $\mathcal{D}_{t+1} \leftarrow \mathcal{D}_{t+1} \cup D_t^k$
14:     **end for**
15:     // ▼ **Ranking Agent**
16:     Evaluate each candidate answer in $\mathcal{A}_t$ using $Q$
17:     Update history context $\mathcal{H}_{t+1} \leftarrow \text{sort}\,(\mathcal{A}_t, \text{key} = Q)$
18: **end for**
19: **Set** $a \leftarrow \text{MajorityVote}(\mathcal{A}_t)$
20: **return** $a$

---

## B. Datasets

We evaluate our framework on seven diverse medical benchmarks, covering a wide spectrum of difficulty, from standardized licensing exams to complex expert-level clinical reasoning. The detailed statistics of these benchmarks are presented in Table 6.

- **MedQA** (Jin et al., 2021) is a standard benchmark for evaluating professional medical knowledge. We utilize the standard test split of the USMLE subset to assess the model's clinical decision-making capability, which is derived from the United States Medical Licensing Examination.

- **MedMCQA** (Pal et al., 2022) is a large-scale dataset collected from Indian medical entrance examination, designed to test general medical knowledge across a broad range of subjects. We evaluate on the validation split of the dataset.

- **Medbullets** (Chen et al., 2025) comprises practice questions sourced from the Medbullets education platform. We utilize the 5-option version to serve as a challenging test set that requires discriminating among multiple plausible distractors.

- **MMLU-Pro** (Wang et al., 2024) is an enhanced and robust version of the massive multitask language understanding benchmark, specifically designed to reduce noise and increase difficulty. We select the health-related subset to assess the model's complex reasoning capabilities.

- **NEJM-AI** (Katz et al., 2024) is derived from official Israeli medical board residency examinations published as part of the New England Journal of Medicine AI collection. It comprises multiple-choice questions from five core clinical specialties and assesses how models handle complex, literature-style medical queries. For NEJM-AI, which contains questions with multiple correct options, a prediction is considered correct only if all ground-truth answer selections are exactly matched.

- **MedExpQA** (Alonso et al., 2024) is a multilingual benchmark for medical question answering grounded in resident medical licensing examinations. We utilize the standard test split of the English version.

*Table 6.* Summary of the statistics and characteristics for the seven medical Q&A benchmarks used in our evaluation. **Size** refers to the number of questions used for evaluation. **Single** and **Multi** denote the presence of a unique correct option versus multiple correct answers.

| Property | MedQA | MedMCQA | Medbullets | MMLU-Pro | NEJM | MedExpQA | MedXpertQA |
|---|---|---|---|---|---|---|---|
| Size | 1273 | 4183 | 308 | 818 | 655 | 125 | 490 |
| # Options | 4 | 4 | 5 | $3 \sim 10$ | 4 | 5 | 10 |
| Answer Mode | Single | Single | Single | Single | Multi | Single | Single |

- **MedXpertQA** (Zuo et al., 2025) is an expert-level benchmark characterized by information-dense queries and high reasoning complexity, simulating real-world clinical decision support scenarios. We use the Text subset and partition the dataset based on the "body system" tag to create a distinct split for training the evaluator model described in Sec. 3.4.

## C. Baseline Methods

This section describes the backbone models and compared baselines in Sec. 4.1.

### C.1. General and Medical LLMs

We first test a range of strong backbone models, including both general-purpose LLMs and specialized medical LLMs, to establish robust baselines.

**General LLMs.** Llama-3.1-8B-Instruct (Grattafiori et al., 2024) is selected as a representative baseline within the 8B parameter class. As a widely adopted open-weight model optimized for instruction following and general reasoning, it serves as a standard benchmark for assessing model capability in the absence of domain-specific medical adaptation. Qwen3-8B and Qwen3-32B (Yang et al., 2025) represent the latest iteration of the Qwen series, distinguished by their superior capabilities in mathematics, coding, and general knowledge. We employ Qwen3-8B as the primary backbone for our comparative evaluation, leveraging its exceptional performance among similarly sized models to test the efficacy of our framework.

**Medical Specialized LLMs.** UltraMedical-3.1-8B (Zhang et al., 2024) is a medical-domain specialized model derived from Llama-3.1-8B. It is trained via supervised instruction tuning on large-scale medical instructions, followed by preference-based optimization to improve medical reasoning and alignment. HuatuoGPT-o1-8B (Chen et al., 2024) is a domain-specific model that adopts a reasoning-oriented training paradigm. It is explicitly optimized based on Llama-3.1-8B-Instruct to perform long-chain reasoning and self-correction for complex medical decision-making tasks.

### C.2. Inference Scaling and RAG Methods

To strictly evaluate the effectiveness of MA-RAG, we compare it against three categories of strong baselines: pure inference-time scaling strategies, naive RAG methods, and representative adaptive RAG frameworks. We also present a simplified workflow diagram to illustrate the overall pipelines of these methods, as shown in Figure 7.

**Inference Scaling Methods without RAG.** These methods rely solely on the model's parametric knowledge, allocating additional test-time computation to enhance reasoning. **Chain-of-Thought (CoT)** (Wei et al., 2022) encourages the model to generate intermediate reasoning steps before deriving the final answer. We utilize a 3-shot prompting strategy in our experiments. **Self-Consistency (SC)** (Wang et al., 2023) is an ensemble strategy that samples diverse reasoning paths and aggregates them via majority voting. To ensure a fair comparison of computational budget with MA-RAG's iterative process, we sample 20 reasoning paths for each question. **Multi-round Refine (Multi-Refine)** (Tian et al., 2025; Xu et al., 2025) employs an iterative self-correction mechanism where the model is prompted to critique and improve its own previous response over multiple rounds. We maintain identical experimental configurations to MA-RAG, excluding only the retrieval module, to isolate the impact of external information. **MDAgents** (Kim et al., 2024) is a multi-agent collaboration framework that automatically assigns solo or group collaboration structures to a team of LLMs tailored to the complexity of the medical task at hand. We implement MDAgents using the same Qwen3-8B backbone for a fair comparison.

**Naive RAG.** These methods incorporate external medical evidence using fixed retrieval patterns. **Single-round RAG (SR-RAG)** (Xiong et al., 2024) represents the standard RAG pipeline where relevant documents are retrieved once based on the initial user query. The model then generates a response conditioned on this static, augmented context. **Fixed-Length**

**RAG (FL-RAG)** (Borgeaud et al., 2022; Ram et al., 2023) is a passive retrieval strategy that unconditionally triggers retrieval at fixed token intervals during the generation process to provide periodic context updates. We set the retrieval trigger interval to 128 tokens. **Fixed-Sentence RAG (FS-RAG)** (Trivedi et al., 2023) operates similarly to FL-RAG but aligns retrieval with syntactic boundaries. It triggers a search operation at the end of every generated sentence to ground the subsequent content.

**Adaptive RAG.** These approaches dynamically determine when to retrieve and dynamically generate queries for retrieval. **FLARE** (Jiang et al., 2023) leverages the model's confidence scores to guide retrieval. It anticipates the upcoming sentence and, if low-confidence tokens are detected, uses the prediction to retrieve relevant documents and regenerate the segment. **TC-RAG** (Jiang et al., 2025) introduces a Turing-complete framework that models RAG as a state-conditioned system. It utilizes a stack-based memory and monitored state variables to autonomously decide when and what to retrieve, enabling adaptive planning and error correction in complex medical scenarios.

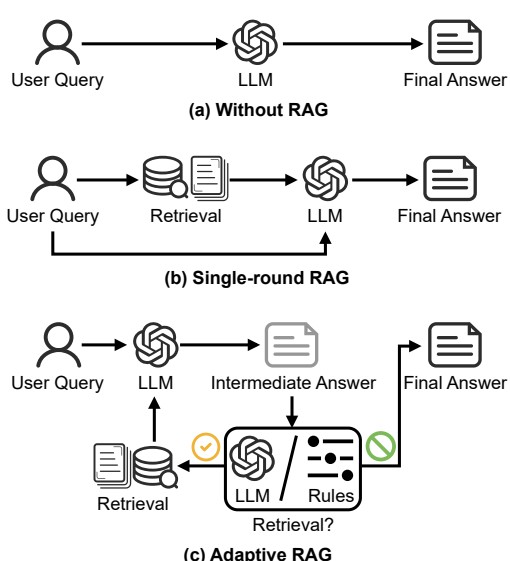

## D. Experimental Implementation

In this section, we provide the detailed implementation settings for the MA-RAG framework, including the prompt templates used for the agents and the specific configurations for the RAG pipeline. The default maximum round for iterative refinement is set to $T = 8$, with generating $K = 4$ queries and retrieving top-8 documents. All experiments with Qwen3 were conducted with the `thinking` mode disabled, except for generating conflict-guided queries.

*Figure 7.* Overall pipelines for different RAG frameworks. (a) **Without RAG**: the model relies solely on its parametric knowledge to generate an answer; (b) **Single-round RAG**: external documents are retrieved once based on the user query and appended as a static context for final answer generation; (c) **Adaptive RAG**: the model dynamically decides when to trigger retrieval during generation and formulates context-aware queries to obtain external evidence, enabling interleaved retrieval and reasoning.

### D.1. Prompt Engineering

We design specific prompts for the solver agent and the retrieval agent to facilitate the iterative refinement process. We use placeholders enclosed in double brackets (e.g., {{variable}}) to denote dynamic content inserted during inference.

**Solver Agent Prompt.** The solver agent is responsible for generating candidate responses. Its input integrates three key components:

1. The task instruction and original medical query and options ($\mathcal{I}_{\text{solver}}$, $q$).
2. The iteratively updated document context ($\mathcal{D}_t$), which provides external evidence.
3. The ranked history of previous reasoning traces ($\mathcal{H}_t$), which serves as in-context demonstrations to guide the model toward higher-quality generation. Explicit scores, from either intrinsic uncertainty or the extrinsic verification, are attached to each history response to indicate its reliability.

**Retrieval Agent Prompt.** The retrieval agent identifies knowledge gaps by analyzing semantic disagreements. It receives the original question and the set of diverse candidate answers generated by the solver agent. The agent is instructed to:

1. Analyze the differences (e.g., conflicting diagnoses, inconsistent symptom interpretations) among the candidates.
2. Abstract these differences into specific information needs.
3. Generate precise search queries optimized for the BM25 retriever to resolve these conflicts.

### D.2. RAG Implementation

We ensure a rigorous and reproducible RAG setup by aligning our infrastructure with established benchmarks. This section provides comprehensive details of the knowledge corpus, the retrieval and re-ranking pipeline, and the precise configurations employed for both the baseline methods and the proposed MA-RAG framework.

*Table 7.* Statistics of the constructed dataset $\mathcal{D}_{qa}$ for training the external evaluator. We report the number of unique questions sourced from each benchmark and the resulting distribution of correct ($y = 1$) and incorrect ($y = 0$) reasoning traces after stratified sampling.

| Metric | MedQA | MedMCQA | MedExpQA | MedXpertQA |
|---|---|---|---|---|
| Unique Questions | 8000 | 4000 | 428 | 1960 |
| Pos. Samples ($y = 1$) | 43604 | 25186 | 2548 | 9777 |
| Neg. Samples ($y = 0$) | 33935 | 16010 | 1783 | 15391 |
| Total Pairs | 77539 | 41196 | 4331 | 25168 |

**Knowledge Corpus.** We utilize MedCorp (Xiong et al., 2024), a large-scale medical retrieval corpus, to ensure a broad coverage of biomedical knowledge. It aggregates high-quality medical texts from diverse sources, including:

- **Textbooks**, a domain-specific collection containing 18 widely used medical textbooks processed into snippets, offering foundational clinical principles and educational knowledge.

- **PubMed**, a comprehensive database of 23.9 million biomedical abstracts, serving as a robust source of scientific evidence and contemporary research findings.

- **StatPearls**, a clinical decision support resource comprising approximately 9,300 articles, providing practical, point-of-care medical guidelines.

- **Wikipedia**, a large-scale general knowledge base consisting of 6.5 million articles, which provides high-level definitions and broad coverage of medical and general concepts.

**Retrieval and Re-ranking Pipeline.** Our retrieval pipeline consists of a sparse first-stage retriever followed by a dense reranker:

- **Retriever (BM25)** (Robertson et al., 2009), a commonly used baseline retriever using bag-of-words and TF-IDF to perform lexical retrieval. We employ the Pyserini implementation to perform an efficient lexical search.

- **Reranker (MedCPT-Cross-Encoder)** (Jin et al., 2023), a lightweight BERT-based model pre-trained on large-scale medical query-article pairs from PubMed search logs. It is employed to re-rank candidate documents based on their fine-grained semantic alignment with the query.

**Retrieval Configuration.** We adopt a standardized two-stage retrieval strategy to balance recall and precision. For any given query, we first utilize BM25 to retrieve the top-32 candidate documents from *each* of the four MedCorp sub-corpora, yielding a diverse initial pool of 128 documents. Subsequently, we select the top-8 documents with the highest semantic relevance scores given by MedCPT reranker to serve as the final external context. In the specific context of MA-RAG, where the agent generates multiple conflict-driven queries (i.e., $K = 4$) in a single refinement round, we retrieve and retain the top-2 documents for each of the 4 queries, ensuring that a total of 8 documents are integrated into the context per round.

## E. Training Details of Extrinsic Evaluator

In Sec. 3.4, we introduced a learned evaluator designed to assess the semantic validity of generated reasoning traces beyond simple statistical uncertainty. This section provides comprehensive details on the training pipeline for this component. We first detail the construction of the query-response dataset used for supervision, followed by the specific architectural choices and hyperparameters employed during fine-tuning.

### E.1. Data Generation Details

To train the external evaluator $V_\theta$ to distinguish between factual and hallucinated reasoning, we constructed a specialized dataset $\mathcal{D}_{qa}$ containing diverse query-response pairs. Table 7 summarizes the statistics of the constructed training dataset.

**Data Sources and Splitting.** We aggregate training questions from four medical benchmarks: MedQA, MedMCQA, MedExpQA, and MedXpertQA. To ensure strict separation between evaluator training and downstream evaluation, we construct the training set exclusively from the official training splits of MedQA, MedMCQA, and MedExpQA. For MedXpertQA, which does not provide a standard training split, we partition the dataset into two non-overlapping subsets using a 4:1 ratio based on the unique `body_system` tags. The larger subset is used for training, while the remaining

subset serves as the held-out test set for the comprehensive experiments. The remaining three benchmarks (Medbullets, MMLU-Pro, and NEJM) are completely unseen during evaluator training, serving as out-of-domain held-out datasets.

**Response Sampling.** We employ Qwen3-8B as the generator model to synthesize "step-by-step thinking" reasoning traces. To promote semantic diversity among candidates, we utilize high-entropy sampling settings with temperature $T = 1.0$, top-$k = 20$, and top-$p = 0.95$. For each query, we generate 128 raw candidate responses to ensure a comprehensive exploration of the reasoning space.

**Labeling and Balancing.** We parse the final predicted option from each generated response and validate it against the ground truth to assign binary correctness labels ($y = 1$ for correct, $y = 0$ for incorrect). To mitigate severe class imbalance and ensure that the verifier learns robust decision boundaries, we employ a stratified sampling strategy. Specifically, for each question, we retain a maximum of 8 distinct correct and 8 distinct incorrect responses, ensuring that the incorrect ones are distributed across different predicted options to maximize error diversity.

### E.2. Training Details

**Model Architecture.** We instantiate the evaluator $V_\theta$ using the pre-trained ModernBERT-base (Warner et al., 2025) architecture, which contains 149M parameters. ModernBERT is selected for its efficiency and ability to handle longer contexts compared to standard BERT models, making it well-suited for processing complex questions and lengthy medical reasoning traces. The input sequence is constructed by concatenating the medical query $q$ and the candidate response $a$, separated by special tokens. We utilize the final hidden state of the `[CLS]` token as the semantic representation of the query-response pair, which is then projected via a linear classification head to produce a scalar logit.

*Table 8.* Hyperparameter settings for fine-tuning the evaluator model.

| Hyperparameter | Value |
|---|---|
| Base Model | ModernBERT-base |
| Optimizer | AdamW |
| Peak Learning Rate | 1e-4 |
| Global Batch Size | 256 |
| Gradient Norm | 1.0 |
| Training Epochs | 10 |
| LR Scheduler | Cosine with Warmup |
| Warmup Ratio | 0.3 |
| Max Sequence Length | 2048 |
| Training Precision | FP16 |

**Optimization Configuration.** The evaluator is optimized as a binary classifier to estimate factual correctness by minimizing the objective defined in Eq. 6. We conduct full-parameter fine-tuning, with specific training hyperparameters listed in Table 8. During inference, the raw logit output $Q_{\text{ext}}(a) = V_\theta(q, a)$ is first normalized to the $[0, 1]$ probability interval via a sigmoid function and subsequently mapped linearly to an integer score in $[0, 10]$, providing a granular quality signal for the ranking agent.

## F. Additional Analysis and Experiments

To further analyze MA-RAG and assess its robustness and generality, we present a set of complementary experiments organized around three analytical perspectives.

**Component Design.** We examine the design choices of MA-RAG's internal components: the impact of query granularity $K$ in conflict-driven retrieval, alternative strategies for enhancing candidate diversity, and a hybrid ranking strategy combining intrinsic and extrinsic signals.

**Competitiveness and Attribution.** We compare MA-RAG against static test-time scaling strategies such as Best-of-$N$ re-ranking, verify that its performance gains do not merely stem from the trained extrinsic evaluator by augmenting baselines with the same evaluator, and demonstrate that MA-RAG is orthogonal to and compatible with existing RAG paradigms.

**Robustness and Efficiency.** We evaluate robustness to sampling variance across independent runs and report an additional inference efficiency analysis on Medbullets.

Unless otherwise specified, all experiments in this section are conducted using the Qwen3-8B backbone with the MA-RAG-ext configuration.

*Table 9.* Performance of MA-RAG-ext under varying query granularities ($K$) using the Qwen3-8B backbone. We vary the number of conflict-driven queries generated per round while maintaining a fixed global retrieval budget of $|\mathcal{D}| = 8$ documents. The best results are highlighted in **bold**.

| $K$ | MedQA | MedMCQA | Medbullets | MMLU-Pro | NEJM | MedExpQA | MedXpertQA | Avg. |
|---|---|---|---|---|---|---|---|---|
| $K=1$ | 75.3 | **67.7** | 60.1 | 70.4 | 58.9 | 76.8 | 20.8 | 61.4 |
| $K=2$ | **77.3** | 67.3 | **60.3** | **70.8** | 58.5 | 76.8 | 21.2 | 61.7 |
| $K=4$ | 77.1 | 67.2 | 59.1 | 70.7 | **60.5** | **78.4** | **22.2** | **62.2** |

*Table 10.* Per-round numerical results of MA-RAG-ext under varying candidate pool sizes $N$, evaluated on Medbullets and MedXpertQA with the Qwen3-8B backbone.

| Dataset | $N$ | Round 1 | Round 2 | Round 3 | Round 4 |
|---|---|---|---|---|---|
| Medbullets | $N=4$ | 50.3 | 56.2 | 55.2 | 57.4 |
| | $N=8$ | 53.2 | 60.4 | 59.1 | 59.1 |
| | $N=16$ | 52.3 | 60.4 | **62.0** | **62.0** |
| MedXpertQA | $N=4$ | 14.7 | 19.2 | 19.0 | 18.6 |
| | $N=8$ | 16.5 | 21.2 | 21.8 | 22.0 |
| | $N=16$ | 16.5 | 21.2 | 23.1 | **23.5** |

## F.1. Impact of Query Granularity in Conflict Resolution

In the retrieval agent, the parameter $K$ controls the granularity of conflict resolution by determining how many distinct queries are generated to address disagreements among candidate answers. To isolate the impact of query diversity from the sheer volume of external information, we conduct an experiment where we vary $K \in \{1, 2, 4\}$ while maintaining a fixed global retrieval budget of $|\mathcal{D}| = 8$ documents per round. Specifically, for $K=1$, we retrieve the top-8 documents for the single query; for $K=2$, the top-4 documents per query; and for $K=4$, the top-2 documents per query.

The results are demonstrated in Table 9, revealing a positive correlation between query granularity and reasoning accuracy. This trend suggests that decomposing complex medical conflicts into multiple, specific search queries allows for broader coverage of the dispute points compared to a single query. This benefit is particularly pronounced on expert-level benchmarks characterized by high reasoning complexity. For instance, on MedXpertQA, increasing $K$ from 1 to 4 yields a monotonic performance improvement, culminating in a performance gain of 1.4 points.

## F.2. Alternative Methods for Candidate Diversity Enhancement

We further investigate two alternative strategies for promoting candidate diversity on MA-RAG-ext: i) **Instruction-Level Perturbation**: synonymously rewriting task instructions fed to the solver agent, and ii) **Advanced Stochastic Sampling**: adopting a more aggressive decoding strategy by setting top-$p=0.99$ and top-$k=40$ to flatten probability distributions.

As shown in Table 11, both alternative strategies yield marginal or inconsistent gains by introducing surface-level perturbations within a constrained candidate pool. In contrast, simply enlarging the candidate pool consistently improves performance (+2.9 on Medbullets, +1.3 on MedXpertQA; see Table 10) by facilitating richer diversity via extensive stochastic sampling.

## F.3. Analysis of Hybrid Ranking and Oracle Upper Bound

We further analyze the effectiveness and limitations of different history context ranking strategies, including a hybrid approach that combines intrinsic and extrinsic signals, as well as an Oracle-based upper bound.

**Hybrid Ranking Strategy.** We first explore a hybrid strategy, denoted as MA-RAG-hyb, which integrates intrinsic entropy with an extrinsic learned evaluator to estimate the quality of candidate reasoning traces. The motivation is that intrinsic uncertainty signals and external verification cues may provide complementary information for ranking history contexts. Specifically, we formulate the composite quality score as $Q_{\text{hyb}} = Q_{\text{ext}} - Q_{\text{int}}$, penalizing high-entropy generations while rewarding high confidence from the learned evaluator. The resulting scores are then linearly scaled to the integer range $[0, 10]$ for integration into the prompt.

**Oracle Ranking Upper Bound.** To better understand the theoretical limit of history context optimization, we further compare MA-RAG with an Oracle ranking strategy. The Oracle utilizes ground-truth answers to construct the optimal

*Table 11.* Comparison of candidate diversity enhancement methods with MA-RAG-ext (Qwen3-8B).

| Method | Medbullets | MedXpertQA |
|---|---|---|
| MA-RAG-ext ($N=8$, $T=8$) | 59.1 | 22.2 |
| + Instruction Perturbation | 60.4 | 20.6 |
| + Stochastic Sampling | 59.7 | 22.9 |
| MA-RAG-ext ($N=16$, $T=4$) | **62.0** | **23.5** |

*Table 12.* Comparison of MA-RAG variants, including intrinsic entropy, extrinsic evaluator, and a hybrid strategy MA-RAG-hyb against the Oracle ranking upper bound on Qwen3-8B. The best results are highlighted in **bold**.

| Method | MedQA | MedMCQA | Medbullets | MMLU-Pro | NEJM | MedExpQA | MedXpertQA | Avg. |
|---|---|---|---|---|---|---|---|---|
| MA-RAG-int | 77.0 | 67.1 | 57.1 | **70.9** | **60.8** | 72.8 | 21.2 | 61.0 |
| MA-RAG-ext | **77.1** | **67.2** | 59.1 | 70.7 | 60.5 | **78.4** | 22.2 | **62.2** |
| MA-RAG-hyb | 76.0 | 66.7 | **61.0** | 69.9 | 59.4 | 74.4 | **22.4** | 61.4 |
| Oracle | 84.5 | 76.0 | 67.5 | 78.0 | 70.2 | 80.0 | 30.9 | 69.6 |

history context $\mathcal{H}_t$. Specifically, at each generation round, the $N$ candidate responses are ranked according to:

- **Correctness**: Responses matching the ground truth are strictly prioritized over incorrect ones.

- **Conciseness**: Among responses with identical correctness, shorter responses are preferred, motivated by the intuition that concise and accurate reasoning traces serve as more effective in-context demonstrations.

As shown in Table 12, the hybrid strategy achieves performance comparable to individual metrics, suggesting that a simple linear combination of uncertainty and verification signals yields limited gains. However, the Oracle's superior performance empirically validates the underlying efficacy of history optimization, confirming that prioritizing high-quality reasoning traces is critical for boosting multi-round generation. The performance gap between MA-RAG and the Oracle suggests that while effective, current quality proxies—entropy and the learned evaluator—have not yet fully captured the optimal ranking. This finding underscores the potential of context optimization and highlights that developing more precise verification metrics remains a key direction for further unlocking the capabilities of test-time scaling.

### F.4. MA-RAG vs. Static Re-ranking (Best-of-N)

To position our iterative refinement approach against standard test-time scaling strategies, we benchmark MA-RAG against simply Best-of-N (BoN) re-ranking. In this experiment, we utilize the 20 reasoning paths generated by Self-Consistency as the candidate pool. We employ both *Intrinsic Entropy* and the *Extrinsic Evaluator* to score and rank these candidates. We report `Recall@k`, defined as the accuracy where the correct answer is present within the top-$k$ ranked candidates. Note that `Recall@1` represents the standard performance of a Best-of-N strategy (i.e., selecting the single highest-scored answer), while `Pass@20` represents the oracle upper bound achievable from the candidate pool.

The results are summarized in Table 13 and visualized in Figure 8. We observe the following key insights:

**Reliability of Scoring Metrics.** Both scoring mechanisms demonstrate effective discriminative capability. Specifically, using the Extrinsic Evaluator for top-1 selection (`Recall@1`) achieves 58.5%, surpassing the unsupervised majority voting of SC (56.7%). This indicates that the learned evaluator is a more robust judge of correctness than simple consensus. Furthermore, the Extrinsic Evaluator consistently outperforms Intrinsic Entropy at `Recall@1` (58.5% vs. 56.7%) and `Recall@2` (63.0% vs. 61.8%), confirming that a verifier trained on semantic correctness provides a cleaner signal than statistical uncertainty.

**The Necessity of Fault Tolerance and Superiority of MA-RAG.** While the scoring metrics are reliable, the significant performance jump from `Recall@1` to `Recall@2` highlights a critical limitation of static selection: the top-1 choice often misses the ground truth, yet the correct answer is frequently latent within the top few candidates. This steep trajectory underscores the necessity of "fault tolerance"—a robust system must leverage the broader information from the top-$k$ pool rather than betting solely on the single highest-scored response. MA-RAG effectively implements this principle by employing the ranking agent to structure the high-quality candidates into a coherent history context. By allowing the solver agent to synthesize these prioritized demonstrations, MA-RAG-ext (62.2%) significantly outperforms the static Best-of-N

*Table 13.* Comparison between static Best-of-N re-ranking and MA-RAG under different scoring mechanisms on Qwen3-8B. We report `Recall@k` to measure whether the correct answer appears in the top-$k$ ranked candidates scored by Intrinsic Entropy or an Extrinsic Evaluator, and include `Pass@20` as the oracle upper bound of the candidate pool.

| Method | MedQA | MedMCQA | Medbullets | MMLU-Pro | NEJM | MedExpQA | MedXpertQA | Avg. |
|---|---|---|---|---|---|---|---|---|
| SC | 73.3 | 62.4 | 51.9 | 66.5 | 56.6 | 70.4 | 15.7 | 56.7 |
| Pass@20 | 90.1 | 83.7 | 78.2 | 82.6 | 77.2 | 90.4 | 40.2 | 77.5 |
| *Intrinsic Entropy* | | | | | | | | |
| Recall@1 | 73.1 | 61.9 | 52.6 | 66.4 | 55.7 | 71.2 | 16.3 | 56.7 |
| Recall@2 | 78.3 | 67.5 | 60.4 | 71.3 | 62.0 | 72.8 | 20.6 | 61.8 |
| MA-RAG-int | 77.0 | 67.1 | 57.1 | 70.9 | 60.8 | 72.8 | 21.2 | 61.0 |
| *Extrinsic Evaluator* | | | | | | | | |
| Recall@1 | 73.7 | 62.0 | 56.8 | 66.3 | 53.2 | 76.8 | 20.4 | 58.5 |
| Recall@2 | 77.6 | 67.7 | 62.3 | 71.6 | 59.8 | 76.8 | 25.3 | 63.0 |
| MA-RAG-ext | 77.1 | 67.2 | 59.1 | 70.7 | 60.5 | 78.4 | 22.2 | 62.2 |

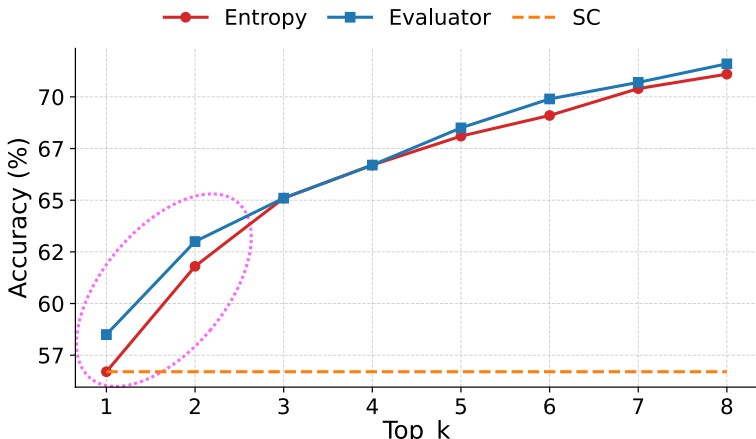

*Figure 8.* Recall@k performance curves for Intrinsic Entropy and Extrinsic Evaluator rankings as $k$ increases from 1 to 8. The circled regions at $k=1$ and $k=2$ highlight the steep improvement in recall, demonstrating that while the top-1 candidate often misses the ground truth, the correct answer is frequently covered within the top-2. This sharp rise validates MA-RAG's design, which leverages the broader context of top-ranked candidates rather than relying on a single static selection.

baseline (58.5%). Essentially, our framework converts the high recall of the top-$k$ ranking into high precision in the final generation, thereby taking a significant step towards the theoretical oracle ceiling of `Pass@20`, as illustrated in Table 13.

### F.5. Augmenting Baselines with the Extrinsic Evaluator

A natural question is whether MA-RAG-ext's performance gains over baselines are attributable to the agentic refinement framework or simply to the trained extrinsic evaluator. To disentangle these factors, we augment representative baselines with the identical extrinsic evaluator and examine whether this closes the gap with MA-RAG-ext. The evaluator is integrated into each baseline as follows:

- **SC**: We directly apply the evaluator to score the 20 reasoning paths sampled in the main results (Table 1) and select the top-ranked response as the final answer.

- **FLARE**: We run FLARE 16 independent times per question and then use the evaluator to select the best answer from the 16 candidates.

- **Multi-Refine**: We integrate the evaluator into the multi-round refinement loop to rank candidates between iterations.

- **TC-RAG**: We do not apply the evaluator to TC-RAG, as its output interleaves the LLM's generated thoughts with retrieved document observations, making the evaluator difficult to apply. We include TC-RAG with SC for comparison.

*Table 14.* Performance of baselines with and without the extrinsic evaluator against MA-RAG-ext on Medbullets and MedXpertQA (Qwen3-8B). The best results are highlighted in **bold**.

| Dataset | Base | | | FLARE | | | TC-RAG | | Multi-Refine | | MA-RAG |
|---|---|---|---|---|---|---|---|---|---|---|---|
| | – | SC | Eval. | – | SC | Eval. | – | SC | – | Eval. | Eval. |
| Medbullets | 51.0 | 51.9 | 56.8 | 51.9 | 52.6 | 54.2 | 49.0 | 53.6 | 55.8 | 56.5 | **59.1** |
| MedXpertQA | 16.1 | 15.7 | 20.4 | 17.7 | 18.2 | 19.6 | 18.0 | 20.6 | 16.3 | 20.4 | **22.2** |

*Table 15.* Compatibility analysis of MA-RAG with standard RAG baselines on Qwen3-8B. We evaluate the impact of combining static initialization (SR-RAG) with dynamic evolution (retrieval agent). Note that the ranking agent is excluded in these experiments to strictly isolate the effects of retrieval strategies. The best results are highlighted in **bold**.

| Method | MedQA | MedMCQA | Medbullets | MMLU-Pro | NEJM | MedExpQA | MedXpertQA | Avg. |
|---|---|---|---|---|---|---|---|---|
| Multi-Refine | 74.7 | 63.6 | 55.8 | 69.1 | 58.0 | 73.6 | 16.3 | 58.7 |
| +Static Context | 74.1 | 68.1 | 55.5 | 69.6 | 61.4 | 72.0 | 18.0 | 59.8 |
| +Dynamic Context | **77.1** | 66.2 | **57.5** | 69.6 | 60.3 | **74.4** | 19.2 | 60.6 |
| +Hybrid Context | 75.3 | **68.6** | 57.1 | **70.0** | **64.0** | 73.6 | **22.2** | **61.5** |

As shown in Table 14, equipping the baselines with the extrinsic evaluator yields consistent improvements over their original counterparts (e.g., +4.7 for SC and +4.1 for Multi-Refine on MedXpertQA), confirming the evaluator's discriminative capability over diverse reasoning trajectories. Nevertheless, MA-RAG-ext consistently maintains a pronounced edge over all augmented counterparts. Specifically, MA-RAG-ext outperforms the strongest baseline, Multi-Refine+Evaluator, by +2.6 points on Medbullets and +1.8 points on MedXpertQA. This performance margin underscores that the success of MA-RAG-ext does not merely stem from a stronger verifier, but fundamentally arises from our agentic loop that dynamically couples conflict-driven evidence with ranked contextual state optimization.

### F.6. Compatibility with Existing RAG Baselines

While MA-RAG operates retrieval at the *inter-round* level—updating the document context between generation cycles via the retrieval agent—standard RAG methods typically operate at the *initialization* stage (e.g., SR-RAG) or *intra-round* level (e.g., TC-RAG). Consequently, our conflict-guided evolution mechanism is orthogonal to, and thus compatible with, existing retrieval paradigms. To demonstrate this flexibility, we evaluate a hybrid configuration where the retrieval agent is integrated on top of a standard Single-Round RAG (SR-RAG) baseline.

In this experiment, we compare four settings using the Qwen3-8B backbone. To isolate the impact of retrieval strategies, we omit the ranking agent across all configurations:

- **No Retrieval** performs iterative refinement relying solely on parametric knowledge, corresponding to the Multi-Refine baseline.

- **Static Context** initializes the document context $\mathcal{D}_1$ with retrieved documents based on the user query. This context remains invariant across all refinement rounds.

- **Dynamic Context** begins with empty document context ($\mathcal{D}_1 = \emptyset$) and iteratively updates $\mathcal{D}_t$ solely by retrieval agent. This mirrors the ablation setting described in Sec. 4.2.

- **Hybrid Context** initializes the document context $\mathcal{D}_1$ with retrieved documents based on the user query to provide a "warm start" and iteratively evolves $\mathcal{D}_t$ by retrieval agent.

The results presented in Table 15 demonstrate the synergistic benefits of this hybrid approach. While Static Context outperforms the No Retrieval baseline (+1.1 points), it is generally surpassed by the Dynamic Context (+1.9 points), suggesting that evolving evidence based on reasoning conflicts is more effective than static, one-shot retrieval. Crucially, the Hybrid Context achieves the highest performance (+2.8 points), combining a "warm start" of a globally relevant initial context with the precision of conflict-driven updates. These findings confirm that the retrieval agent effectively complements traditional RAG pipelines, serving as a robust "plug-and-play" module to enhance reasoning fidelity and consistency.

*Table 16.* Robustness analysis across 4 independent inference runs on 4 challenging benchmarks with Qwen3-8B. The best results are highlighted in **bold**.

| Method | Medbullets | MMLU-Pro | NEJM | MedXpertQA |
|---|---|---|---|---|
| FLARE | 51.6±1.1 | 61.7±0.4 | 55.6±0.4 | 17.6±1.2 |
| TC-RAG | 49.1±1.1 | 64.8±1.0 | 57.2±0.7 | 18.5±1.1 |
| MDAgents | 52.2±1.7 | 65.7±0.9 | 57.1±0.5 | 18.2±1.0 |
| MA-RAG-ext | **59.4±0.6** | **70.5±0.7** | **60.8±0.3** | **22.0±0.6** |

*Table 17.* Inference efficiency comparison across methods on Medbullets with Qwen3-8B. We report average retrieved documents, generated tokens, final answer tokens, and wall-clock time per question.

| Method | Avg. Docs | Avg. Gen. Tokens | Avg. Final Tokens | Avg. Time (s) | Accuracy |
|---|---|---|---|---|---|
| Base (Qwen3-8B) | 0 | 449 | 449 | 5.6 | 51.0 |
| Self-Consistency (SC) | 0 | 9,016 | 451 | 22.0 | 51.9 |
| FLARE | 32.2 | 767 | 377 | 32.2 | 51.9 |
| TC-RAG | 12.9 | 1,229 | 1,177 | 49.0 | 49.0 |
| MDAgents | 8.0 | 5,810 | – | 115.4 | 52.2 |
| MA-RAG-ext | 9.0 | 8,810 | 496 | 41.1 | **59.1** |

## F.7. Robustness to Sampling Variance

To assess the stability and reproducibility of MA-RAG under inherent stochastic sampling (e.g., temperature, top-$k$, top-$p$), we evaluate the performance variance across 4 independent inference runs on 4 challenging benchmarks. We compare MA-RAG-ext against three competitive baselines—FLARE, TC-RAG, and MDAgents—under identical experimental configurations with the Qwen3-8B backbone. Table 16 reports the mean accuracy and standard deviation for each method.

The results demonstrate that MA-RAG-ext achieves the highest mean accuracy across all four benchmarks, corroborating its overall superiority. More importantly, MA-RAG-ext shows markedly lower variance than baselines, indicating that semantic conflict serves as a more stable retrieval signal compared to the token-level uncertainty metrics used by FLARE and TC-RAG.

## F.8. Inference Efficiency Analysis on Medbullets

To complement the efficiency analysis on MedXpertQA in Sec. 4.5, we report an additional inference efficiency of MA-RAG-ext against representative baselines on Medbullets. Table 17 summarizes the results across retrieval, generation, and wall-clock time dimensions using the Qwen3-8B backbone under identical hardware conditions.

Consistent with the MedXpertQA findings, MA-RAG-ext retrieves fewer documents than FLARE (9.0 vs. 32.2) yet achieves substantially higher accuracy, and outperforms MDAgents by 6.9 points while requiring only about one-third of its inference time (41.1s vs. 115.4s). These results reinforce that semantic conflict provides a more efficient retrieval signal than token-level uncertainty, and MA-RAG achieves a favorable cost-performance trade-off.

# G. Validation of Conflict-Driven Query Generation

A core premise of MA-RAG is that semantic conflict among candidate responses can serve as a reliable signal for pinpointing knowledge gaps and guiding targeted retrieval. To validate this premise, we conduct a comprehensive evaluation combining a quantitative LLM-as-a-Judge assessment with qualitative case studies, directly examining whether the LLM can accurately diagnose genuine medical disputes and formulate actionable retrieval queries.

## G.1. Quantitative Evaluation via LLM-as-a-Judge

Since obtaining human-annotated ground truth for conflicts among diverse candidate responses is highly challenging, we leverage an LLM-as-a-Judge approach to assess the generated queries across three dimensions:

- **Faithfulness** ($\in \{0, 1\}$): Measures whether all generated queries accurately reflect a factual contradiction genuinely present in the candidate responses. This serves as the core reliability metric. Intuitively, maintaining high Faithfulness becomes more challenging as the number of generated queries ($K$) increases.

*Table 18.* Multi-model LLM-as-a-Judge evaluation of conflict-driven queries generated by MA-RAG on Medbullets (500 samples). Three advanced models independently score the queries across Faithfulness, Clinical Relevance, and Comprehensiveness of Coverage under varying query granularity $K$.

| Model | $K$ | Faithfulness | Relevance | Comprehensiveness |
|---|---|---|---|---|
| | $K=1$ | 0.98 | 4.20 | 2.71 |
| Qwen3.6 Plus | $K=2$ | 0.96 | 4.22 | 3.26 |
| | $K=4$ | 0.84 | 4.19 | 3.73 |
| | $K=1$ | 0.93 | 4.61 | 1.44 |
| DeepSeek V3.2 | $K=2$ | 0.90 | 4.50 | 2.16 |
| | $K=4$ | 0.83 | 4.72 | 3.69 |
| | $K=1$ | 0.99 | 4.08 | 2.97 |
| MiniMax M2.7 | $K=2$ | 0.93 | 4.01 | 3.16 |
| | $K=4$ | 0.85 | 4.19 | 3.82 |

- **Clinical Relevance** ($\in [1, 5]$): Evaluates how critical the targeted conflicts are to the actual medical diagnosis or patient outcome, distinguishing core medical disagreements from trivial wording differences.

- **Comprehensiveness of Coverage** ($\in [1, 5]$): Assesses whether the generated queries collectively capture the full spectrum of distinct medical disagreements among the candidates. This value is expected to naturally increase with $K$, as a larger set of queries can successfully identify more diverse and independent points of contention.

We employ three advanced models—Qwen3.6 Plus, DeepSeek V3.2, and MiniMax M2.7—to score the queries generated by our Qwen3-8B backbone on 100 randomly sampled questions from the Medbullets dataset. The average scores are presented in Table 18, where $K$ corresponds to the query granularity in the retrieval agent.

These findings highlight the 8B model's robust capability in diagnosing genuine medical disputes. Evaluated across multiple advanced models, the generated queries consistently maintain high Faithfulness (exceeding 83% even at $K=4$) alongside strong Clinical Relevance (averaging above 4.0 across all settings). Moreover, the scoring trends align precisely with expectations: scaling the query granularity $K$ significantly enhances the Comprehensiveness of Coverage, while Faithfulness exhibits a marginal, anticipated decrease reflecting the growing complexity of formulating multiple distinct, high-quality queries. The strong inter-model agreement among the three models further reinforces the reliability of this assessment.

In summary, these quantitative results firmly validate the precision of the extracted conflicts and the high quality of the generated queries, demonstrating that the 8B model can reliably diagnose genuine medical disputes and translate them into actionable retrieval queries.

### G.2. Qualitative Case Studies

To further illustrate how MA-RAG diagnoses genuine medical disputes and transforms them into actionable retrieval queries, we present two representative case studies sampled from Medbullets and MedQA.

---

**Case Study of Clinical Dispute Resolution from Medbullets**

**Question:** A 65-year-old man with a history of HIV (CD4 count 150 cells/$\mu$L) presents for a routine visit. He received the 13-valent and 23-valent pneumococcal vaccines 15 years ago, and a second 23-valent dose 10 years ago. In addition to encouraging ART compliance, which prophylactic medication and vaccine are indicated?

**Candidate Responses:**

- **Candidate 1:** "…the **pneumococcal vaccines are not routinely recommended** for administration in patients with HIV …**the zoster vaccine may be considered** in this patient … The correct answer is E (Trimethoprim-sulfamethoxazole and zoster vaccine)." **(Incorrect)**

- **Candidate 2:** "…the CDC recommends that patients with HIV receive the **pneumococcal vaccine** …Since this patient has a CD4 count of 150 cells/$\mu$L, the **zoster vaccine is not currently indicated** …The correct answer is C (Trimethoprim-sulfamethoxazole and pneumococcal vaccine)." **(Correct)**

**Generated Queries:**

1. pneumococcal vaccine cd4 150 hiv indication
2. zoster vaccine cd4 150 hiv contraindication

---

**Case Study of Mutual Hallucination Diagnosis from MedQA**

**Question:** A 47-year-old female undergoes a thyroidectomy for treatment of Graves' disease. Post-operatively, she reports a hoarse voice and difficulty speaking. You suspect that this is likely a complication of her recent surgery. What is the embryologic origin of the damaged nerve that is most likely causing this patient's hoarseness?

**Candidate Responses:**

- **Candidate 1:** "...which gives rise to the recurrent laryngeal nerve, originates from the **3rd and 4th pharyngeal arches** ..." **(Incorrect)**
- **Candidate 2:** "...the closest embryologic correlate to the RLN is the **1st pharyngeal arch** ..." **(Incorrect)**

**Generated Query:**

1. Which pharyngeal arch is responsible for the recurrent laryngeal nerve

---

In summary, these case studies illustrate MA-RAG's central premise: semantic conflict among candidate responses serves as an effective and interpretable signal for diagnosing knowledge gaps, and the retrieval agent can reliably translate such conflicts into actionable queries that bridge those gaps through targeted evidence acquisition.

## H. Limitations

Although MA-RAG achieves strong performance gains and a favorable cost-performance trade-off, its multi-round agentic refinement incurs additional inference-time computation that may still be prohibitive for real-time clinical applications with stringent latency requirements. The effectiveness of conflict-guided retrieval is also constrained by the coverage and quality of the underlying medical corpus. Broader resources such as clinical case repositories, web-scale medical search, and structured knowledge bases can be further incorporated. Moreover, the optimization of history context depends on answer quality estimation signals that remain imperfect. While the trained evaluator yields promising gains, developing more accurate, robust, and lightweight evaluation metrics could further unlock the potential of iterative refinement.

**Prompt Template for Solver Agent (Round=1)**

You are a medical assistant, please answer my medical questions. Give your final choice (capital option) closed in tag <**answer**>**X**</**answer**> after your analysis.
Your response should be as detailed as possible, but please do not use any subheadings.

Below is a multiple-choice question.
**Question**
{{question}}

**Options**
{{options}}

Please analyze and then re-answer. Please give your response towards **higher score**. Give your analysis and final answer.

**Prompt Template for Solver Agent (Round>1)**

You are a medical assistant, please answer my medical questions. Give your final choice (capital option) closed in tag <**answer**>**X**</**answer**> after your analysis.
Your response should be as detailed as possible, but please do not use any subheadings.

Below is a multiple-choice question.
**Question**
{{question}}

**Options**
{{options}}

**Documents**
{{documents}}

- - -

I will provide several assistant's previous answers and their scores (0-10) for your reference (higher scores indicate higher quality). But these previous answers may be incorrect, please analyze and then re-answer.

1. The previous answer is (Score: x):
{{answers_1}}

...

N. The previous answer is (Score: x):
{{answers_N}}

- - -

The above are the question along with the assistant's previous answers, and some relevant documents. Please analyze and then re-answer. Please give your response towards **higher score**. Give your analysis and final answer.

---

**Prompt Template for Retrieval Agent**

**Role & Goal:**
You are a medical expert. I will provide multiple different answers to the same question, which may be incorrect. Your task is to identify contradictions, ambiguities, and core dispute points among the answers, and extract key concept queries for further retrieval and verification.

**Processing Steps**:
1. Analyze different answers and summarize the differences between them.
2. Extract keywords from these differences for retrieval.
3. Generate 4 precise queries for further search using **BM25** retriever.

**Output format**:
```
[Query 1] xxx
[Query 2] xxx
[Query 3] xxx
[Query 4] xxx
```

Below is a multiple-choice question.
**Question**
{{question}}

**Options**
{{options}}

- - -

I will provide several assistant's previous answers, which may be incorrect:

1. The previous answer is:
{{answers_1}}

...

N. The previous answer is:
{{answers_N}}

- - -

Please analyze the differences among the answers and generate 4 queries for further search.

