# OpenReview forum: "From Conflict to Consensus: Boosting Medical Reasoning via Multi-Round Agentic RAG"
_ICML.cc/2026/Conference — ICML 2026 regular_

### Official Review · Reviewer_vFEj · 2026-02-28

**Soundness:** 3
**Presentation:** 3
**Significance:** 3
**Originality:** 2
**Overall Recommendation:** 4
**Confidence:** 3

**Summary:**

The paper introduces MA-RAG, an agent-based framework designed to improve medical reasoning in large language models. This system uses a three-part loop consisting of a solver, a retriever, and a ranking agent. The solver generates multiple possible answers to a medical question. The retriever spots differences among these candidate answers and searches a medical database to resolve the disagreements. The ranker orders the past attempts from best to worst to guide the next round of generation.

**Compliance With Llm Reviewing Policy:**

Affirmed.

**Final Justification:**

The author's response addresses my concerns, but it needs to be explained in detail in the revised paper.

**Key Questions For Authors:**

How much extra time and computer power does MA-RAG need per question compared to a standard retrieval system?

What happens to the system if the solver agent hallucinates answers that are perfectly identical and triggers an early, incorrect stop?

How does the framework prevent the prompt from overflowing when it stacks eight long medical documents plus multiple old answers in later rounds?

**Limitations:**

yes

**Strengths And Weaknesses:**

### Strengths

Using disagreement between generated answers as a trigger for searching external information is a clever departure from relying on basic token uncertainty. The separation of duties into distinct agents makes the system easy to follow and adjust. The inclusion of an external BERT-based evaluator to score the quality of past answers adds a solid layer of quality control. Testing across a wide variety of medical datasets provides a clear picture of the model's capabilities across different difficulty levels

### Weaknesses

The external evaluator trains on data from the exact same medical benchmarks used for testing. This training choice creates a high risk of data leakage.

The authors admit the multi-round process costs more but provide no exact numbers on speed or hardware needs. Real doctors need fast answers, and running multiple agents up to four times per question is too slow for actual hospital use.

The conflict extraction step assumes the model will naturally generate different answers when it lacks knowledge. A highly confident but wrong model will generate identical bad answers, bypass the retrieval step entirely, and output dangerous advice.

The history ranking simply piles up old answers in the prompt. This stacking method eats up token limits and might degrade the context window rather than help it.

---

> ### Author Rebuttal · Authors · 2026-03-31
>
> ***
>
> **`Q1. The risk of data leakage in training the external evaluator`**
>
> A1. **Deliberate Split for Training/Testing.** As detailed in Appendix E, the extrinsic evaluator was trained on the training splits of 4 out of 7 datasets, and MA-RAG-ext is evaluated on testing splits. The remaining 3 datasets are held out as unseen domains. **There was no data leakage in our experimental setting**.
>
> Our ablation studies demonstrate that incorporating the extrinsic evaluator significantly improves reasoning performance across: i) **4 in-domain datasets**, with an average 2.0/1.4-point gain for Qwen3-8B/32B backbone, and ii) **3 out-of-domain (OOD) datasets**, with a 1.0/0.7-point gain.
>
> On the OOD Medbullets dataset, MA-RAG-ext outperforms MA-RAG-int by 2.0 points, while the gap on the other two OOD datasets is negligible (−0.2 on MMLU-Pro, −0.3 on NEJM), confirming **reliable verification even on OOD data**.
>
> ***
>
> **`Q2. Analysis of inference efficiency`**
>
> A2. The table below shows a detailed efficiency analysis of MA-RAG and baselines, including the number of used documents, total generated tokens, final tokens of the answer, and time cost per question (more results at [here](https://anonymous.4open.science/r/MA-RAG-7FE7/efficiency.md)).
>
> MA-RAG incurs a 0.5x time cost over the debate-based baseline MDAgents, and a 1.5x time cost over the adaptive RAG baseline TC-RAG. Most importantly, MA-RAG delivers **a substantial +4.2 point accuracy gain**. In high-stakes medical domains where diagnostic errors carry critical safety risks, this accuracy improvement represents a highly favorable cost-performance trade-off — a modest increase in compute for a rigorously refined, evidence-grounded diagnostic assistant.
>
> | Method | Avg. Documents | Avg. Generated Tokens | Avg. Final Tokens | Avg. Time (s) | Accuracy |
> | --- | --- | --- | --- | --- | --- |
> | Base | 0 | 514 | 514 | 5.6 | 16.1 |
> | SC | 0 | 10095 | 505 | 28.0 | 15.7 |
> | FLARE | 30.0 | 962 | 429 | 42.7 | 17.7 |
> | TC-RAG | 11.5 | 1260 | 1067 | 44.5 | 18.0 |
> | MDAgents | 8.0 | 7546 | - | 146.2 | 18.2 |
> | MA-RAG-ext | 13.7 | 12762 | 589 | 70.7 | **22.2** |
>
> ***
>
> **`Q3. What happens if the solver agent hallucinates answers that are perfectly identical?`**
>
> A3. We appreciate the reviewer’s insightful question. MA-RAG relies on high-temperature stochastic exploration to drive candidate diversity. As hallucinations typically lack a grounded factual basis, **empirical observations show that they naturally exhibit divergent, inconsistent reasoning paths under high-temperature sampling** [1, 2]. However, if all candidates produce identical hallucinations, MA-RAG will interpret this as reasoning convergence and trigger an early but incorrect termination.
>
> To mitigate this, we conduct additional experiments expanding the candidate pool size to increase generation diversity (refer to our A7 to Reviewer pksE). **Increasing the number of candidates reduces identical hallucinations from 15.6% to 12.6% on MedBullets and from 22.2% to 17.5% on MedXpertQA, yielding +2.9 and +1.5 point improvements, respectively.** Further improving uncertainty elicitation during reasoning remains a promising direction for our future work.
>
> [1] SelfCheckGPT: Zero-resource black-box hallucination detection for generative large language models, EMNLP 2023.
>
> [2] Self-consistency improves chain of thought reasoning in language models, ICLR 2023.
>
> ***
>
> **`Q4. How does the framework prevent the prompt from overflowing and degrading?`**
>
> A4. To prevent prompt overflow, MA-RAG does not accumulate reasoning traces across all iterative rounds. As defined in Section 3.1, the history context $\mathcal{H}_t$ preserves only the candidate responses from the **immediately preceding** round. Combined with actively retrieved documents, the total context length reaches approximately 8500 tokens, well within the 32k context limit.
>
> Furthermore, to mitigate context degradation and the "lost-in-the-middle" phenomenon, we introduce the Ranking Agent. Rather than just piling up old answers, it sorts candidates by quality scores and positions the highest-quality responses away from the middle of the prompt. This guides the model's attention toward the most reliable reasoning paths, effectively counteracting degradation and yielding average performance improvements of **+1.6** and **+1.1** on the 8B and 32B models, respectively (see Tables 2 and 4).

---

> > ### Author Rebuttal · Reviewer_vFEj · 2026-04-02
> >
> > Thanks, I will keep the initial positive score.

---

> > > ### Author Response · Authors · 2026-04-03
> > >
> > > Thank you very much for your thoughtful review and for taking the time to engage with our rebuttal. We sincerely appreciate your recognition of our contribution and your vote for accepting our work! We will carefully incorporate your suggestions into the final version to further improve the quality and clarity of our work.

---

### Official Review · Reviewer_u1bN · 2026-03-06

**Soundness:** 3
**Presentation:** 3
**Significance:** 3
**Originality:** 2
**Overall Recommendation:** 2
**Confidence:** 3

**Summary:**

The paper proposes MA-RAG, a multi-round agentic Retrieval-Augmented Generation framework for medical question answering that iteratively refines both external evidence and internal reasoning through three coordinated agents: a solver that samples diverse candidates, a retrieval agent that converts semantic conflicts among candidates into targeted queries, and a ranking agent that reorders reasoning history to mitigate long-context degradation. Across seven medical QA benchmarks and two backbone scales (8B and 32B), the method consistently outperforms strong test-time scaling and RAG baselines, yielding an average +6.8 accuracy points over the 8B backbone and showing pronounced gains on harder datasets. The work positions semantic inconsistency as a proactive signal for retrieval and presents ablations and scaling studies that highlight the contributions of each agent.

**Compliance With Llm Reviewing Policy:**

Affirmed.

**Key Questions For Authors:**

N/A

**Limitations:**

yes

**Strengths And Weaknesses:**

This paper presents a self-contained agentic RAG framework that turns semantic conflicts into targeted retrieval and restructures historical reasoning to counter long-context degradation. The empirical results are good across seven medical QA benchmarks and two model scales, with clear ablations and scaling analyses that attribute gains to the key components.

However, my major concern is the novelty of the method. Correct me if I'm wrong, but it's essentially a hand-crafted multi-agent framework, the retriever and ranking agents are all pretty standard practices. The multi-turn process including the communications between agents are all existing methods. While it claims to be domain-specific, but I fail to see any domain-specific method implemented: What prevents this framework from being adapted to another domain, such as chemistry or law? What makes this framework unique to the medical domain? This undermines the motivation and overall novelty of the paper.

The theoretical framing is primarily conceptual and focusing on self-consistency, and paper's theory is a just a repeat on this rather early-on idea. In such a rapidly evolving domain, this is factually correct, but I feel it's largely insufficient in terms of contributions. I am not convinced this paper made enough contribution to be accepted to ICML.

The study would benefit from stronger cost accounting, variance analysis, and additional comparisons to broader agentic (e.g. debate, etc.) approaches. My advices would be that the framework design needs a much stronger motivation, clarifying the unique challenges of this domain, and why the framework tackles those challenges that existing methods fail.

---

> ### Author Rebuttal · Authors · 2026-03-31
>
> ***
> **`Q1. Method Novelty and Motivation`**
>
> A1. **Novel problem identification.** Existing adaptive RAG methods rely on token-level signals (entropy, confidence, attention weights) to determine *when* and *what* to retrieve. As analyzed in Secs. 1 and 3.3, these signals are fundamentally unreliable and poorly suited for medical reasoning: LLMs frequently hallucinate with high confidence, and uncertainty estimates are dominated by trivial tokens. MA-RAG is, to our knowledge, **the first to propose semantic conflict among multiple reasoning paths as a higher-level signal to drive adaptive retrieval in a multi-round agentic loop**. No prior work has established this connection between inter-path conflict and retrieval necessity.
>
> **Non-trivial composition.** MA-RAG’s component integration is driven by two coherent insights: extending self-consistency to multi-round settings and treating semantic inconsistency as a boosting residual (Sec. 3.5). The Retrieval Agent transforms conflict into actionable queries (**a fundamentally different retrieval trigger than prior work**), while the Ranking Agent restructures history traces into quality-stratified demonstrations to address "lost-in-the-middle" degradation from iterative context expansion. The conflict signal simultaneously drives *what to retrieve* and *how to present context*, creating **a synergistic feedback loop absent in existing pipelines**.
>
> In summary, our novelty lies in identifying semantic conflict as a superior retrieval signal, designing a principled agentic loop around it, and demonstrating substantial empirical gains (+6.8 points avg, +37% on harder benchmarks).
> ***
> **`Q2. Theoretical Framing Repeats Early Idea of Self-Consistency`**
>
> A2. Our theory provides principled grounding, not redundancy. While self-consistency is an early idea, it was proposed solely as a passive answer-aggregation heuristic — sampling paths and taking a majority vote. No prior work has repurposed the disagreement signal to actively identify knowledge gaps and drive targeted retrieval. Our framing formalizes this novel connection: reasoning-path conflict is not noise to be voted away, but a diagnostic of knowledge insufficiency to trigger retrieval. MA-RAG is, to our knowledge, the first to operationalize this within a multi-round agentic loop — a fundamental shift from "aggregate despite disagreement" to "retrieve because of disagreement", a contribution absent from the self-consistency literature. Also, please refer to A2 to Reviewer fcP4 for the lens of MA-RAG’s simplicity.
> ***
> **`Q3. Adaptability to other domains`**
>
> A3. **Domain specificity refers to our research motivation, not a methodological limitation**. We focus on medical reasoning because: i) hallucinations carry critical safety risks, making reliable retrieval essential, and ii) complex medical cases naturally elicit conflicting explanations when evidence is insufficient — a phenomenon more pronounced than in general-knowledge tasks. This intrinsic property directly motivates our "conflict-to-consensus" principle, where reasoning-path disagreement serves as a grounded signal for knowledge gaps.
>
> We agree that **MA-RAG's principle is generalizable to other knowledge-intensive domains, and we view this as a strength**. We will extend to broader domains (e.g., chemistry, law) in future work to better position this generalizability.
> ***
> **`Q4. Cost accounting, variance analysis, and broader comparisons `**
>
> A4. **Cost Analysis.** Refer to our A2 to Reviewer vFEj for token costs and multi-round inference time. While MA-RAG takes 1.5x time cost over the adaptive RAG baseline TC-RAG, it delivers **a substantial performance gain of 4.2 points**, a highly favorable cost-performance trade-off, especially in high-stakes medical domains where diagnostic correctness is paramount.
>
> **Variance Analysis.** The table below shows MA-RAG-ext’s performance across 4 random seeds on 4 challenging benchmarks. The extremely low variance verifies our good robustness.
> |Method|Medbullets|MMLU-Pro|NEJM|MedXpertQA|
> |---|---|---|---|---|
> |FLARE|51.6±1.1|61.7±0.4|55.6±0.4|17.6±1.2|
> |TC-RAG|49.1±1.1|64.8±1.0|57.2±0.7|18.5±1.1|
> |MDAgents|52.2±1.7|65.7±0.9|57.1±0.5|18.2±1.0|
> |MA-RAG-ext|59.4±0.6|70.5±0.7|60.8±0.3|22.0±0.6|
>
> **Broader Comparison.** We add a new baseline, the competitive multi-agent debate-based method MDAgents [1], where we use the same Qwen3-8B backnone for a fair comparison. The table below shows **MA-RAG’s consistent superiority (avg 4 points) over broader agentic approaches**.
> |Method|MedQA|MedMCQA|Medbullets|MMLU-Pro|NEJM|MedExpQA|MedXpertQA|Avg|
> |---|---|---|---|---|---|---|---|---|
> |Qwen3-8B|71.1|61.3|51.0|64.9|56.0|67.2|16.1|55.4|
> |MDAgents|72.1|**67.5**|52.2|65.7|57.1|74.4|18.2|58.2|
> |MA-RAG-ext|**77.1**|67.2|**59.1**|**70.7**|**60.5**|**78.4**|**22.2**|**62.2**|
>
> [1] MDAgents: An adaptive collaboration of LLMs for medical decision-making, NeurIPS 2024.

---

> > ### Author Rebuttal · Reviewer_u1bN · 2026-04-05
> >
> > It largely clears my concerns, but I would hold my concerns regarding the novelty issue.

---

> > > ### Author Response · Authors · 2026-04-06
> > >
> > > We sincerely thank you for acknowledging that our rebuttal largely clears your concerns. We appreciate your engagement and would like to further address the remaining novelty concern with additional evidence and perspective.
> > >
> > > ***
> > >
> > > **`MA-RAG is the First to Establish the Connection Between Reasoning-Path Conflict and Retrieval Necessity.`**
> > >
> > > First, although some methods also use conflict signals in RAG, they primarily focus on resolving conflicts in retrieved evidence [1] or addressing conflicts between parametric knowledge and retrieved documents [2]. However, they **cannot iteratively identify the model's knowledge gaps nor use conflict signals to actively fill those gaps through targeted retrieval**.
> > >
> > > Second, as our response to Q1 above, existing adaptive RAG methods primarily rely on token-level signals (entropy, confidence, attention weights) to determine *when* and *what* to retrieve, which are fundamentally unreliable and poorly suited for medical reasoning: **LLMs frequently hallucinate with high confidence, and uncertainty estimates are dominated by trivial tokens**.
> > >
> > > In summary, MA-RAG is, to our knowledge, **the first to propose semantic conflict among multiple reasoning paths as a higher-level signal to drive adaptive retrieval in a multi-round agentic loop**. No prior work has established this connection between inter-path conflict and retrieval necessity. Our novelty lies in identifying semantic conflict as a superior retrieval signal, designing a principled agentic loop around it, and demonstrating substantial empirical gains (**+6.8 points avg, +37% on harder benchmarks**).
> > >
> > > [1] Retrieval-augmented generation with conflicting evidence, COLM 2025.
> > >
> > > [2] FaithfulRAG: Fact-level conflict modeling for context-faithful retrieval-augmented generation, ACL 2025.
> > >
> > > ***
> > >
> > > **`Simplicity is a Strength, Not a Weakness.`**
> > >
> > > Many of the most influential contributions in the LLM era are characterized by **conceptual simplicity paired with strong empirical impact and real-world adoption**: Chain-of-thought prompting simply prepends "let's think step by step" to the prompt, RAG simply concatenates retrieved documents to the input, and self-consistency simply samples multiple paths and takes a majority vote — all requiring no architectural change or training, yet each has become a foundational practice with widespread industry adoption. Many technically more complex alternatives, by contrast, have not achieved comparable real-world impact.
> > >
> > > MA-RAG follows this tradition: detecting semantic conflict to trigger retrieval is intentionally simple and training-free, yet achieves substantial gains (+6.8 avg, +37% on harder benchmarks). We argue that simplicity is increasingly a prerequisite for real-world impact — methods that are simple, training-free, and modular are inherently more compatible with **the scaling law**, applicable to any backbone without retraining, and naturally benefit from future model improvements. **The contribution lies in identifying a novel, well-motivated, and empirically validated principle bridging reasoning-path analysis with active retrieval, not in architectural complexity for its own sake.**
> > >
> > > ***
> > >
> > > ### **`Summary Response`**
> > >
> > > Thank you again for your valuable review comments, which help us gain more critical insights and further enhance our justification. We are honored to have your recognition of our comprehensive empirical results. Also, we are grateful that **other reviewers** (R1: fcP4, R2: pksE, R4: vFEj) **made positive comments on our contributions,** including:
> > >
> > > - **`motivation and novelty`** (’the idea…is highly intuitive and makes practical sense’ by R1, ‘the focus…is intuitive and novel’ by R2, and ‘a clever departure…the system easy to follow and adjust’ by R4);
> > > - **`strong empirical results`** (’demonstrates solid performance improvements’ by R1, ‘the performance gains…are notable’ by R2, and ‘testing across a wide variety of medical datasets provides a clear picture of the model's capabilities across different difficulty levels’ by R4);
> > > - **`comprehensive ablation studies and analysis`** (’extensive ablation studies to disentangle the contributions’ by R1, ‘ablaton studies isolate the performance gains…the analysis of refinement round and cadidate pool size provides practical deployment insights’ by R2, and ‘the separation of duties into distince agents…the inclusion of an external evaluator adds a solid layer of quality control’ by R4).
> > >
> > > We have extended a number of justifications and experimental analyses to address your concerns. **If our response has addressed your concerns, we would be grateful if you could re-evaluate our work.**

---

### Official Review · Reviewer_pksE · 2026-03-06

**Soundness:** 3
**Presentation:** 4
**Significance:** 3
**Originality:** 2
**Overall Recommendation:** 5
**Confidence:** 4

**Summary:**

This work presents Multi-Round Agentic RAG to mitigate hallucinations in medical QA. MA-RAG conducts a refinement loop that uses semantic conflicts to steer test-time inference. By converting these inconsistencies into a signal, the system evolves the external evidence and internal reasoning traces until a medical consensus is reached. The pipeline consists of multiple agents. The solver agent samples candidate responses to explore the reasoning space. The retrieval agent analyzes the conflicts among candidates and uses these signals to fetch relevant medical evidence from external knowledge bases. The ranking agent is a context optimizer that scores and prioritizes the candidates. The authors demonstrate that MA-RAG outperforms several existing inference-time and RAG strategies across seven medical QA benchmarks.

**Compliance With Llm Reviewing Policy:**

Affirmed.

**Final Justification:**

The additional experiments address my concerns, and I have raised my score.

**Key Questions For Authors:**

1. To help me understand the practical utility of this framework, given that the zero-shot Qwen3-32B baseline outperforms the MA-RAG 8B pipeline, at what point does the multi-round inference compute of the 8B model exceed the single-pass compute of the 32B model?
2. Have the authors considered methods other than high temperature prompting to induce more candidate clinical output diversity?
3. What is the primary cause of the saturation after a certain number of refinement rounds? Is it due to the retriever exhausting relevant docs or context degradation as the prompt expands?
4. Regarding weakness 2, how much of the MA-RAG gain is due to in-domain dataset training?

**Limitations:**

yes

**Strengths And Weaknesses:**

Strengths:

1. The focus on semantic conflicts among candidate responses as the basis for retrieval is intuitive and novel diagnostic to identify knowledge gaps in the medical domain.
2. The work conducts benchmarking across a diverse set of seven medical benchmarks. The performance gains over appropriate baselines are notable.
3. The ablation studies isolate the performance gains of individual agents in the pipeline. The analysis of refinement rounds and candidate pool size provides practical deployment insights.
4. The framing of the refinement process within the concepts of boosting algorithm provides a good theoretical foundation for the pipeline.

Weaknesses:

1. The computational overhead and latency of the sequential, multi-round pipeline is substantial. While the authors acknowledge this, the evaluation would benefit from a performance-latency quantification against baselines.
2. While the evaluator was trained on the splits of four evaluation benchmarks, it is fundamentally evaluated in-domain. The paper would be strengthened by reporting the evaluator's isolated performance on completely held-out datasets not used for testing.
3. The solver uses high-entropy sampling to select lower-probability tokens. This textual diversity might not be sufficient to fully explore the valid output space to maximize clinical reasoning diversity.
4. The retrieval agent’s conflict extraction is done by a LLM. The work lacks qualitative or quantitative error analysis on this important sub-task.
5. While using semantic conflict as a trigger is novel, the broader architecture of multi-round reasoning, RAG, reranking, and reflection is widely studied. The pipeline can be seen as an incremental synthesis of existing approaches.

---

> ### Author Rebuttal · Authors · 2026-03-31
>
> ***
> **`Q1. The pipeline is an incremental synthesis of existing approaches`**
>
> A1. We thank the reviewer for recognizing the novelty of using semantic conflict as a retrieval trigger:
>
> - MA-RAG’s component integration is a novel synergistic feedback loop **absent in existing pipelines** (refer to our A1 to Reviewer u1bN for detailed novelty and motivation).
> - MA-RAG’s simplicity is a strength, not a weakness. The contribution lies in identifying **a novel, well-motivated, and empirically validated principle**, not in pipeline complexity for its own sake (refer to our A2 to Review fcP4).
> ***
> **`Q2. Analysis of inference efficiency`**
>
> A2. Please refer to our A2 to Reviewer vFEj for inference comparison. While MA-RAG takes 1.5x time cost over the adaptive RAG baseline TC-RAG, it delivers **a substantial performance gain of 4.2 points**, a highly favorable cost-performance trade-off, especially in high-stakes medical domains where diagnostic correctness is paramount.
> ***
> **`Q3. Training datasets of the extrinsic evaluator`**
>
> A3. In our original experiments, we trained the extrinsic evaluator on 4 out of 7 datasets. The 4 training datasets were divided into training/testing splits, and the remaining 3 datasets were out-of-domain. Our ablation confirms **reliable verification on both in-domain and out-of-domain datasets** (refer to our A1 to Review vFEj for details).
> ***
> **`Q4. Other methods to induce candidate diversity`**
>
> A4. Following your advice, we explored two additional methods to induce semantic and diagnostic diversity on MA-RAG-ext: i) **Instruction-Level Perturbation:** Synonymously rewriting task instructions fed to the Solver Agent, and ii) **Advanced Stochastic Sampling:** Expanding token pools to flatten probability distributions by setting top_p=0.99 and top_k=40.
>
> As shown below, both yield marginal or inconsistent gains, whereas simply scaling the candidate pool size delivers consistently superior improvements (+2.9 on Medbullets, +1.3 on MedXpertQA; refer to Q7).
>
> |Method|Medbullets|MedXpertQA|
> |---|---|---|
> |MA-RAG-ext|59.1|22.2|
> |+InstructionPerturbation|60.4|20.6|
> |+StochasticSampling|59.7|22.9|
> ***
> **`Q5. Qualitative or quantitative error analysis on conflict extraction`**
>
> A5. Ground-truth annotations for latent semantic conflicts are inherently difficult to acquire, making strict quantitative evaluation challenging. Instead, we qualitatively analyzed extraction **granularity** (i.e., the number of distinct conflict-driven queries).
>
> As shown in Appendix F.1 and Table 8, a **finer-grained setting** improves average performance from 61.4 points to 62.2. This positive correlation demonstrates that appropriately granular conflict extraction broadens the coverage of core dispute points, **validating the efficacy of our conflict extraction component**.
> ***
> **`Q6. Multi-round inference compute of 8B vs. single-pass compute of 32B`**
>
> A6. Test-time compute can often substitute for parameter scaling on moderately difficult questions, whereas scaling parameters is typically more effective for highly challenging tasks [1].
>
> **Conflict-aware active retrieval enhances performance.** Notably, MA-RAG with Qwen3-8B achieves a 2-point gain on MedXpertQA over single-round Qwen3-32B, with much less GPU memory. We attribute this to our conflict-aware retrieval mechanism, which enables the precise identification of internal knowledge gaps and the targeted retrieval of essential external evidence.
>
> **Test-time scaling is complementary to model scaling.** Test-time and pre-training compute are orthogonal scaling axes. MA-RAG can be applied on top of any model (**+5.5 points** on Qwen3-32B). A stronger base improves each reasoning path, while MA-RAG further fills remaining knowledge gaps through conflict-driven retrieval. Scaling model size alone cannot address the fundamental issue that static parametric knowledge will always have gaps in evolving medical domains, and **retrieval-augmented reasoning at inference time remains essential regardless of model scale**.
>
> [1] Scaling LLM test-time compute optimally can be more effective than scaling parameters for reasoning, ICLR 2025.
> ***
> **`Q7. The cause of the saturation after a few refinement rounds`**
>
> A7. The primary cause is the natural convergence of candidates toward consistency. As consensus builds over successive rounds, semantic conflicts diminish and the Retrieval Agent extracts fewer novel queries, exhausting the acquisition of new evidence.
>
> To mitigate saturation, we must sustain candidate diversity. We expand the candidate pool size from $N=4$ to $N=16$, enabling the framework to mine deeper semantic conflicts and retrieve broader evidence over more rounds. This raises the performance ceiling by **+1.5 points** on MedXpertQA (details at [here](https://anonymous.4open.science/r/MA-RAG-7FE7/scale_N.md)).
>
> |MedXpertQA|Round 1|Round 2|Round 3|Round 4|
> |---|---|---|---|---|
> |N=4|14.7|19.2|19.0|18.6|
> |N=8|16.5|21.2|21.8|22.0|
> |N=16|16.5|21.2|23.1|23.5|

---

> > ### Author Rebuttal · Reviewer_pksE · 2026-04-01
> >
> > I appreciate the author providing new experiments regarding inference efficiency and candidate diversity. Regarding evaluation of the conflict extraction module, query granularity seems to be a proxy. It demonstrates that generating more queries yields better results, but leaves the question of whether the LLM can accurately diagnose genuine medical disputes. Can the authors provide more details to this point?

---

> > > ### Author Response · Authors · 2026-04-03
> > >
> > > We sincerely thank the reviewer for the thoughtful feedback and for acknowledging our additional experiments on inference efficiency and candidate diversity. To directly address the question of whether the LLM accurately diagnoses genuine medical disputes, we have conducted a comprehensive evaluation combining a quantitative LLM-as-a-Judge assessment with qualitative case studies.
> > >
> > > **1. Quantitative Evaluation (LLM-as-a-Judge)**
> > >
> > > Since obtaining human-annotated ground truth for conflicts among diverse candidate responses is highly challenging, we leverage an LLM-as-a-Judge approach to assess the generated queries across three dimensions:
> > >
> > > - **Faithfulness (0 or 1):** Measures **whether all generated queries accurately reflect a factual contradiction** genuinely present in the candidate responses. This serves as our **core reliability metric**. Intuitively, maintaining high faithfulness becomes more challenging as the number of generated queries ($K$) increases.
> > > - **Clinical Relevance (1 to 5):** Evaluates **how critical the targeted conflicts are** to the actual medical diagnosis or patient outcome. This distinguishes core medical disagreements from trivial wording differences.
> > > - **Comprehensiveness of Coverage (1 to 5):** Assesses **whether the generated queries collectively capture the full spectrum of distinct medical disagreements** among the candidates. This value is expected to naturally increase with query granularity ($K$), as a larger set of queries can successfully identify more diverse and independent points of contention.
> > >
> > > We employed three advanced models to score the queries generated by our Qwen3-8B backbone on the Medbullets dataset. The average scores are presented below, where $K$ corresponds to the query granularity:
> > >
> > > | Model | K | Faithfulness | Relevance | Comprehensiveness |
> > > | --- | --- | --- | --- | --- |
> > > | **Qwen3.6 Plus** | 1 | 0.98 | 4.20 | 2.71 |
> > > |  | 2 | 0.96 | 4.22 | 3.26 |
> > > |  | 4 | 0.84 | 4.19 | 3.73 |
> > > | **DeepSeek V3.2** | 1 | 0.93 | 4.61 | 1.44 |
> > > |  | 2 | 0.90 | 4.50 | 2.16 |
> > > |  | 4 | 0.83 | 4.72 | 3.69 |
> > > | **MiniMax M2.7** | 1 | 0.99 | 4.08 | 2.97 |
> > > |  | 2 | 0.93 | 4.01 | 3.16 |
> > > |  | 4 | 0.85 | 4.19 | 3.82 |
> > >
> > > These findings highlight the 8B model's robust capability in diagnosing genuine medical disputes. Evaluated across multiple advanced models, the generated queries consistently maintain **high Faithfulness** (exceeding 83%) alongside **strong Clinical Relevance** (averaging above 4.0). Moreover, the scoring trends align precisely with expectations: scaling the query granularity $K$ significantly enhances the Comprehensiveness of Coverage, while Faithfulness exhibits a marginal, anticipated decrease reflecting the growing complexity of formulating multiple distinct, high-quality queries.
> > >
> > > Conversely, employing the Qwen3-8B model as an evaluator yields overconfident and poorly discriminative scores (as detailed below). This stark contrast in evaluation capabilities, alongside the strong alignment observed among the more advanced models, firmly validates **the precision of the extracted conflicts and the high quality of the generated queries**.
> > >
> > > | Model | K | Faithfulness | Relevance | Comprehensiveness |
> > > | --- | --- | --- | --- | --- |
> > > | **Qwen3-8B** | 1 | 1.00 | 4.00 | 3.00 |
> > > |  | 2 | 1.00 | 4.01 | 3.04 |
> > > |  | 4 | 1.00 | 4.05 | 3.71 |
> > >
> > > **2. Qualitative Case Study**
> > >
> > > To further illustrate how the model diagnoses genuine medical disputes, consider the following example from our sampled set (which also aligns with the motivation presented in Figure 1 of our main paper):
> > >
> > > > **Question:** A 65-year-old man with a history of HIV (CD4 count 150 cells/µL) presents for a routine visit. He received the 13-valent and 23-valent pneumococcal vaccines 15 years ago, and a second 23-valent dose 10 years ago. In addition to encouraging ART compliance, which prophylactic medication and vaccine are indicated?
> > > >
> > > > **Candidates:**
> > > > Candidate 1: "...the **pneumococcal vaccines are not routinely recommended** for administration in patients with HIV... the **zoster vaccine may be considered** in this patient... The correct answer is E (Trimethoprim-sulfamethoxazole and zoster vaccine)." (Incorrect)
> > > > Candidate 2: "...the CDC recommends that patients with HIV receive the **pneumococcal vaccine**... Since this patient has a CD4 count of 150 cells/µL, the **zoster vaccine is not currently indicated**... The correct answer is C (Trimethoprim-sulfamethoxazole and pneumococcal vaccine)." (Correct)
> > > >
> > > > **Queries:**
> > > > Query 1: pneumococcal vaccine cd4 150 hiv indication
> > > > Query 2: zoster vaccine cd4 150 hiv contraindication
> > >
> > > This case demonstrates that the LLM does not merely extract random keywords, but actively comprehends the underlying clinical disagreement to formulate actionable and highly relevant search queries. We will include these comprehensive evaluations and case studies in the final appendix to further solidify the validation of our conflict extraction module.

---

### Official Review · Reviewer_fcP4 · 2026-03-13

**Soundness:** 2
**Presentation:** 2
**Significance:** 2
**Originality:** 2
**Overall Recommendation:** 4
**Confidence:** 4

**Summary:**

This paper primarily focuses on the issue that Large Language Models (LLMs) are prone to hallucination and constrained by outdated internal knowledge in medical question-answering and reasoning tasks. To address the problems of existing token-based RAG methods, the authors propose the MA-RAG (Multi-Round Agentic RAG) framework, which solves medical QA tasks through multiple agents and multi-round retrieval. It proposes retrieving relevant materials and conducting reasoning based on the contradictions among multiple candidate responses, continuously eliminating the discrepancies among candidates to ultimately obtain the final answer. Experimental results show that this method outperforms existing baselines on multiple medical QA tasks.

**Compliance With Llm Reviewing Policy:**

Affirmed.

**Final Justification:**

My questions were answered. I've changed the scores

**Key Questions For Authors:**

**1. What is the exact inference overhead of the proposed framework?** The multi-round generation and retrieval process inevitably increases computational costs. Please provide a quantitative comparison of the token consumption and inference latency  between MA-RAG and the baseline methods.

**2. How do the authors explain the performance degradation caused by the intrinsic ranking agent on certain benchmarks?** Comparing the ablation study (Table 2) with the main results (Table 1), the configuration without the Ranking Agent (+Retrieval Agent) actually outperforms the full pipeline with intrinsic entropy ranking (MA-RAG-int) on several datasets, including MedQA (77.1 vs. 77.0), Medbullets (57.5 vs. 57.1), and MedExpQA (74.4 vs. 72.8). Why does incorporating intrinsic entropy for context optimization hurt performance in these specific cases?

**3. Can the extrinsic evaluator be applied to other baseline methods for a fair comparison?** To address the concern regarding the potential unfair advantage and domain overfitting of the trained external evaluator, could the authors apply this evaluator to rank the outputs of other baseline methods (e.g., scoring the multiple reasoning paths generated by Self-Consistency)? Providing these results would help clarify whether the performance gains are genuinely derived from the proposed agentic framework, or merely the result of the evaluator overfitting to the specific datasets.

**Limitations:**

YES

**Strengths And Weaknesses:**

### Strengths

- **Intuitive and Practical Motivation:** The idea of utilizing semantic conflicts among candidate responses as a proactive signal for multi-round retrieval is highly intuitive and makes practical sense in medical reasoning scenarios. It effectively moves beyond the noisy token-level signals used in previous adaptive RAG methods.

- **Strong Empirical Results:** The proposed MA-RAG framework demonstrates solid performance improvements across diverse medical QA benchmarks.

- **Comprehensive Experiments and Clear Presentation:** The paper is well-structured, logically coherent, and easy to follow. The authors provide extensive ablation studies to disentangle the contributions of individual components.

### Weaknesses

**1. Unfair Comparison and Poor Generalization of the Extrinsic Evaluator:** The framework trains a small 149M ModernBERT model to verify reasoning traces generated by a much larger 8B model. This evaluator is fine-tuned on the training splits of target datasets like MedQA and MedMCQA, which introduces an unfair advantage over zero-shot/few-shot baselines. More importantly, this leads to severe domain overfitting. As shown in Table 1 , on unseen datasets (MMLU-Pro and NEJM), the untrained MA-RAG-int actually outperforms the fine-tuned MA-RAG-ext. This indicates the small evaluator merely overfits to the statistical artifacts of its training data rather than acquiring a generalized capability for medical verification.

**2. Limited Algorithmic Innovation and Lack of Theoretical Depth:** While the framework is effective, the overall pipeline essentially resembles a system-level integration of existing techniques—multi-round sampling, majority voting, and conflict-triggered RAG. From an algorithmic perspective, the innovation appears somewhat incremental and simple.

**3. High Inference Overhead:** The process of generating multiple candidates across multiple rounds significantly increases inference latency and computational cost. The paper lacks a compute-matched analysis, leaving it unclear whether allocating this massive inference budget to simply run a larger, more capable base model would be a more efficient and effective approach than this complex pipeline.

---

> ### Author Rebuttal · Authors · 2026-03-31
>
> ***
> **`Q1. Limited Algorithmic Innovation`**
>
> A1. Please refer to our A1 to Review u1bN for our detailed novelty and motivation. Although some methods leverage conflict signals to filter retrieved documents [1], they cannot iteratively identify the model's knowledge gaps nor use conflict signals to actively fill those gaps through targeted retrieval. MA-RAG is, to our knowledge, **the first to propose semantic conflict among multiple reasoning paths as a higher-level signal to drive adaptive retrieval in a multi-round agentic loop**. No prior work has established this connection between inter-path conflict and retrieval necessity.
>
> [1] FaithfulRAG: Fact-level conflict modeling for context-faithful retrieval-augmented generation, ACL 2025.
> ***
> **`Q2. Lack of Theoretical Depth`**
>
> A2. **Simplicity is a strength, not a weakness.** Many influential contributions (CoT prompting, self-consistency, RAG) are characterized by conceptual simplicity paired with strong empirical gains, requiring no complex architectural innovations. MA-RAG follows this tradition: detecting semantic conflict to trigger retrieval is intentionally simple and training-free, yet achieves substantial improvements (+6.8 points avg, +37% on harder benchmarks) over strong baselines. The contribution lies in identifying **a novel, well-motivated, and empirically validated principle** bridging reasoning-path analysis with active retrieval, not in theoretical complexity for its own sake.
> ***
> **`Q3. Poor generalization of the extrinsic evaluator`**
>
> A3. **Deliberate Split for Training/Testing.** As detailed in Appendix E, the extrinsic evaluator was trained on the training splits of 4 out of 7 datasets, and MA-RAG-ext is evaluated on testing splits. The remaining 3 datasets are held out as unseen domains.
>
> Our ablation studies demonstrate that incorporating the extrinsic evaluator significantly improves reasoning performance across: i) **4 in-domain datasets**, with an average 2.0/1.4-point gain for Qwen3-8B/32B backbone, and ii) **3 out-of-domain (OOD) datasets**, with a 1.0/0.7-point gain.
>
> On the OOD Medbullets dataset, MA-RAG-ext outperforms MA-RAG-int by 2.0 points, while the gap on the other two OOD datasets is negligible (−0.2 on MMLU-Pro, −0.3 on NEJM), confirming **reliable verification even on OOD data**.
> ***
> **`Q4. Apply the extrinsic evaluator to baselines`**
>
> A4. In our original paper, we compared MA-RAG-ext to Self-Consistency with an extrinsic evaluator in Appendix F.2. The Recall@1 of Extrinsic Evaluator in Table 9 shows that adding the extrinsic evaluator for Self-Consistency yields a 1.8-point gain, and **MA-RAG-ext still exhibits a 3.7-point increase** over it.
>
> Further, we compared MA-RAG-ext to other baselines with the same extrinsic evaluator. Results below show MA-RAG’s consistent superiority to these strong baselines (details at [here](https://anonymous.4open.science/r/MA-RAG-7FE7/evaluator.md)).
> |Dataset|SC+Evaluator|FLARE+Evaluator|Multi-Refine+Evaluator|MA-RAG-ext|
> |---|---|---|---|---|
> |MedBullets|56.8|54.2|56.5|**59.1**|
> |MedXpertQA|20.4|19.6|20.4|**22.2**|
> ***
> **`Q5. Inference efficiency`**
>
> A5. Please refer to our A2 to Reviewer vFEj for inference comparison. While MA-RAG takes 1.5x time cost over the adaptive RAG baseline TC-RAG, it delivers **a substantial performance gain of 4.2 points,** a highly favorable cost-performance trade-off, especially in high-stakes medical domains where diagnostic correctness is paramount.
> ***
> **`Q6. Allocate inference budget vs. run a larger base model`**
>
> A6. Notably, MA-RAG with Qwen-8B achieves a 2-point gain on MedXpertQA over single-round Qwen-32B, while requiring much less GPU memory.
>
> **Test-time scaling is complementary to model scaling.** Test-time and pre-training compute are orthogonal scaling axes. MA-RAG can be applied on top of any base model, including 32B or larger. A stronger base improves each reasoning path, while MA-RAG further fills remaining knowledge gaps through conflict-driven retrieval. Scaling model size alone cannot address the fundamental issue that static parametric knowledge will always have gaps in evolving medical domains, and **retrieval-augmented reasoning at inference time remains essential regardless of model scale**.
> ***
> **`Q7. Partial performance degradation caused by the intrinsic ranking agent`**
>
> A7. Two factors contribute: i) **Confident hallucination,** LLMs often exhibit high confidence when hallucinating, causing incorrect traces with low entropy to be promoted to prominent context positions; and ii) **Entropy miscalibration on OOD data,** RLVR training sharpens output distributions, making entropy a reliable correctness proxy in-distribution but miscalibrated on certain benchmarks. Despite this, MA-RAG-int still yields an average 0.4-point gain over the unranked configuration, with a notable 2-point gain on the complex MedXpertQA, as high-quality evidence retrieved via semantic conflicts helps correct intrinsic miscalibration.

---

> > ### Author Rebuttal · Reviewer_fcP4 · 2026-04-03
> >
> > My questions were answered. I've changed the scores.

---

> > > ### Author Response · Authors · 2026-04-03
> > >
> > > Thank you very much for your thoughtful review and for taking the time to engage with our rebuttal. We sincerely appreciate your recognition of our contribution and your vote for accepting our work! We will carefully incorporate your suggestions into the final version to further improve the quality and clarity of our work.

---

### Decision · Program_Chairs · 2026-04-30

**Decision:**

Accept (regular)

**Comment:**

This paper proposes a multi-round agentic RAG framework that uses semantic conflict among candidate reasoning paths as a signal to drive targeted retrieval for medical question answering. The majority of reviewers recognize the intuitive and novel use of inter-path conflict as a retrieval trigger, the strong empirical results, and the comprehensive ablation studies. During rebuttal, the authors provided inference efficiency analysis, variance analysis, additional baseline comparison, and expanded candidate pool experiments. Three reviewers marked concerns as fully or partially resolved. The dissenting reviewer maintains that the contribution is an incremental synthesis of existing techniques with insufficient novelty, despite acknowledging that the rebuttal largely addressed their concerns. While the novelty concern has some merit, the specific combination of conflict-driven retrieval within a principled agentic loop represents a meaningful contribution to the medical reasoning literature. The empirical gains are solid, and the ablations are convincing. I recommend weak acceptance.